# On Deep Generative Models for Approximation and Estimation of Distributions on Manifolds

**Biraj Dahal**[*]
School of Mathematics
Georgia Institute of Technology
bdahal6@gatech.edu

**Alex Havrilla**[*]
School of Mathematics
Georgia Institute of Technology
ahavrilla3@gatech.edu

**Minshuo Chen**
Electrical and Computer Engineering
Princeton University
mc0750@princeton.edu

**Tuo Zhao**
Industrial and Systems Engineering
Georgia Institute of Technology
tourzhao@gatech.edu

**Wenjing Liao**
School of Mathematics
Georgia Institute of Technology
wliao60@gatech.edu

## Abstract

Generative networks have experienced great empirical successes in distribution learning. Many existing experiments have demonstrated that generative networks can generate high-dimensional complex data from a low-dimensional easy-to-sample distribution. However, this phenomenon can not be justified by existing theories. The widely held manifold hypothesis speculates that real-world data sets, such as natural images and signals, exhibit low-dimensional geometric structures. In this paper, we take such low-dimensional data structures into consideration by assuming that data distributions are supported on a low-dimensional manifold. We prove statistical guarantees of generative networks under the Wasserstein-1 loss. We show that the Wasserstein-1 loss converges to zero at a fast rate depending on the intrinsic dimension instead of the ambient data dimension. Our theory leverages the low-dimensional geometric structures in data sets and justifies the practical power of generative networks. We require no smoothness assumptions on the data distribution which is desirable in practice.

## 1 Introduction

Deep generative models, such as generative adversarial networks (GANs) [Goodfellow et al., 2014, Arjovsky et al., 2017] and variational autoencoder [Kingma and Welling, 2013, Mohamed and Wierstra, 2014], utilize neural networks to generate new samples which follow the same distribution as the training data. They have been successful in many applications including producing photorealistic images, improving astronomical images, and modding video games [Reed et al., 2016, Ledig et al., 2017, Schawinski et al., 2017, Brock et al., 2018, Volz et al., 2018, Radford et al., 2015, Salimans et al., 2016].

To estimate a data distribution $Q$, generative models solve the following optimization problem

$$\min_{g_\theta \in \mathcal{G}} \texttt{discrepancy}((g_\theta)_\sharp \rho, Q), \tag{1}$$

---

[*]These authors contributed equally to this work.

36th Conference on Neural Information Processing Systems (NeurIPS 2022).

where $\rho$ is an easy-to-sample distribution, $\mathcal{G}$ is a class of generating functions, `discrepancy` is some distance function between distributions, and $(g_\theta)_\sharp \rho$ denotes the pushforward measure of $\rho$ under $g_\theta$. In particular, when we obtain a sample $z$ from $\rho$, we let $g_\theta(z)$ be the generated sample, whose distribution follows $(g_\theta)_\sharp \rho$.

There are many choices of the `discrepancy` function in literature among which Wasserstein distance attracts much attention. The so-called Wasserstein generative models [Arjovsky et al., 2017] consider the Wasserstein-1 distance defined as

$$W_1(\mu, \nu) = \sup_{f \in \mathrm{Lip}_1(\mathbb{R}^D)} \mathbb{E}_{X \sim \mu}[f(X)] - \mathbb{E}_{Y \sim \nu}[f(Y)], \tag{2}$$

where $\mu, \nu$ are two distributions and $\mathrm{Lip}_1(\mathbb{R}^D)$ consists of 1-Lipschitz functions on $\mathbb{R}^D$. The formulation in (2) is known as the Kantorovich-Rubinstein dual form of Wasserstein-1 distance and can be viewed as an integral probability metric [Müller, 1997].

In deep generative models, the function class $\mathcal{G}$ is often parameterized by a deep neural network class $\mathcal{G}_{\mathrm{NN}}$. Functions in $\mathcal{G}_{\mathrm{NN}}$ can be written in the following compositional form

$$g_\theta(x) = W_L \cdot \sigma(W_{L-1} \ldots \sigma(W_1 x + b_1) + \ldots + b_{L-1}) + b_L, \tag{3}$$

where the $W_i$'s and $b_i$'s are weight matrices and intercepts/biases of corresponding dimensions, respectively, and $\sigma$ is ReLU activation applied entry-wise: $\sigma(a) = \max(a, 0)$. Here $\theta = \{W_i, b_i\}_{i=1}^L$ denotes the set of parameters.

Solving (1) is prohibitive in practice, as we only have access to a finite collection of samples, $X_1, \ldots, X_n \overset{\mathrm{iid}}{\sim} Q$. Replacing $Q$ by its empirical counterpart $Q_n = \frac{1}{n} \sum_{i=1}^n \delta_{X_i}$, we end up with

$$\hat{g}_n = \operatorname*{argmin}_{g_\theta \in \mathcal{G}_{\mathrm{NN}}} W_1((g_\theta)_\sharp \rho, Q_n). \tag{4}$$

Note that (4) is also known as training deep generative models under the Wasserstein loss in existing deep learning literature [Frogner et al., 2015, Genevay et al., 2018]. It has exhibited remarkable ability in learning complex distributions in high dimensions, even though existing theories cannot fully explain such empirical successes. In literature, statistical theories of deep generative models have been studied in Arora et al. [2017], Zhang et al. [2017], Jiang et al. [2018], Bai et al. [2018], Liang [2017, 2018], Uppal et al. [2019], Chen et al. [2020], Lu and Lu [2020], Block et al. [2021], Luise et al. [2020], Schreuder et al. [2021]. Due to the well-known curse of dimensionality, the sample complexity in Liang [2017], Uppal et al. [2019], Chen et al. [2020], Lu and Lu [2020] grows exponentially with respect to underlying the data dimension. For example, the CIFAR-10 dataset consists of $32 \times 32$ RGB images. Roughly speaking, to learn this data distribution with accuracy $\epsilon$, the sample size is required to be $\epsilon^{-D}$ where $D = 32 \times 32 \times 3 = 3072$ is the data dimension. Setting $\epsilon = 0.1$ requires $10^{3072}$ samples. However, GANs have been successful with $60,000$ training samples [Goodfellow et al., 2014].

A common belief to explain the aforementioned gap between theory and practice is that practical data sets exhibit low-dimensional intrinsic structures. For example, many image patches are generated from the same pattern by some transformations, such as rotation, translation, and skeleton. Such a generating mechanism induces a small number of intrinsic parameters. It is plausible to model these data as samples near a low dimensional manifold [Tenenbaum et al., 2000, Roweis and Saul, 2000, Peyré, 2009, Coifman et al., 2005].

To justify that deep generative models can adapt to low-dimensional structures in data sets, this paper focuses (from a theoretical perspective) on the following fundamental questions of both distribution approximation and estimation:

**Q1:** *Can deep generative models approximate a distribution on a low-dimensional manifold by representing it as the pushforward measure of a low-dimensional easy-to-sample distribution?*

**Q2:** *If the representation in Q1 can be learned by deep generative models, what is the statistical rate of convergence in terms of the sample size $n$?*

This paper provides positive answers to these questions. We consider data distributions supported on a $d$-dimensional compact Riemannian manifold $\mathcal{M}$ isometrically embedded in $\mathbb{R}^D$. The easy-to-sample distribution $\rho$ is uniform on $(0, 1)^{d+1}$. To answer **Q1**, our Theorem 1 proves that deep generative

models are capable of approximating a transportation map which maps the low-dimensional uniform distribution $\rho$ to a large class of data distributions on $\mathcal{M}$. To answer **Q2**, our Theorem 2 shows that the Wasserstein-1 loss in distribution learning converges to zero *at a fast rate depending on the intrinsic dimension $d$* instead of the data dimension $D$. In particular we prove that

$$\mathbb{E}W_1((\hat{g}_n)_\sharp\rho, Q) \leqslant Cn^{-\frac{1}{d+\delta}}$$

for all $\delta > 0$ where $C$ is a constant independent of $n$ and $D$.

Our proof proceeds by constructing an oracle transportation map $g^*$ such that $g^*_\sharp\rho = Q$. This construction crucially relies on a cover of the manifold by geodesic balls, such that the data distribution $Q$ is decomposed as the sum of local distributions supported on these geodesic balls. Each local distribution is then transported onto lower dimensional sets in $\mathbb{R}^d$ from which we can apply optimal transport theory. We then argue that the oracle $g^*$ can be efficiently approximated by deep neural networks.

We make minimal assumptions on the network, only requiring that $g_\theta$ belongs to a neural network class (labelled $\mathcal{G}_{\text{NN}}$) with size depending on some accuracy $\epsilon$. Further, we make minimal assumptions on the data distribution $Q$, only requiring that it admits a density that is upper and lower bounded. Standard technical assumptions are made on the manifold $\mathcal{M}$.

## 2 Preliminaries

We establish some notation and preliminaries on Riemannian geometry and optimal transport theory before presenting our proof.

**Notation.** For $x \in \mathbb{R}^d$, $\|x\|$ is the Euclidean norm, unless otherwise specified. $B_X(0, r)$ is the open ball of radius $r$ in the metric space $X$. If unspecified, we denote $B(0, r) = B_{\mathbb{R}^d}(0, r)$. For a function $f : \mathbb{R}^d \to \mathbb{R}^d$ and $A \subseteq \mathbb{R}^d$, $f^{-1}(A)$ denotes the pre-image of $A$ under $f$. $\partial$ denotes the differential operator. For $0 < \alpha \leqslant 1$, we denote by $C^\alpha$ the class of Hölder continuous functions with Hölder index $\alpha$. $\| \cdot \|_\infty$ denotes the $\infty$ norm of a function, vector, or matrix (considered as a vector). For any positive integer $N \in \mathbb{N}$, we denote by $[N]$ the set $\{1, 2, \ldots, N\}$.

### 2.1 Riemannian Geometry

Let $(\mathcal{M}, g)$ be a $d$-dimensional compact Riemannian manifold isometrically embedded in $\mathbb{R}^D$. Roughly speaking a manifold is a set which is locally Euclidean i.e. there exists a function $\phi$ continuously mapping a small patch on $\mathcal{M}$ into Euclidean space. This can be formalized with *open sets* and *charts*. At each point $x \in \mathcal{M}$ we have a *tangent space* $T_x\mathcal{M}$ which, for a manifold embedded in $\mathbb{R}^D$, is the $d$-dimensional plane tangent to the manifold at $x$. We say $\mathcal{M}$ is Riemannian because it is equipped with a smooth metric $g_x : T_x\mathcal{M} \times T_x\mathcal{M} \to \mathbb{R}$ (where $x$ is a basepoint) which can be thought of as a local inner product. We can define the Riemannian distance $d_\mathcal{M} : \mathcal{M} \times \mathcal{M} \to \mathbb{R}$ on $\mathcal{M}$ as

$$d_\mathcal{M}(x, y) = \inf\{L(\gamma)|\gamma \text{ is a } C^1(\mathcal{M}) \text{ curve such that } \gamma(0) = x, \gamma(1) = y\},$$

i.e. the length of the shortest path or *geodesic* connecting $x$ and $y$. An *isometric embedding* of the $d$-dimensional $\mathcal{M}$ in $\mathbb{R}^D$ is an embedding that preserves the Riemannian metric of $\mathcal{M}$, including the Riemannian distance. For more rigorous statements, see the classic reference Flaherty and do Carmo [2013].

We next define the exponential map at a point $x \in \mathcal{M}$ going from the tangent space to the manifold.

**Definition 1** (Exponential map). *Let $x \in \mathcal{M}$. For all tangent vectors $v \in T_x\mathcal{M}$, there is a unique geodesic $\gamma$ that starts at $x$ with initial tangent vector $v$, i.e. $\gamma(0) = x$ and $\gamma'(0) = v$. The exponential map centered at $x$ is given by $\exp_x(v) = \gamma(1)$, for all $v \in T_x\mathcal{M}$.*

The exponential map takes a vector $v$ on the tangent space $T_x\mathcal{M}$ as input. The output, $\exp_x(v)$, is the point on the manifold obtained by travelling along a geodesic curve that starts at $x$ and has initial direction $v$ (see Figure 1 for an example).

It is well known that for all $x \in \mathcal{M}$, there exists a radius $\delta$ such that the exponential map restricted to $B_{T_x\mathcal{M}}(0, \delta)$ is a diffeomorphism onto its image, i.e. it is a smooth map with smooth inverse. As

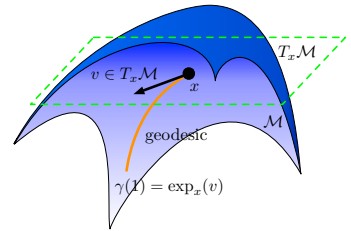
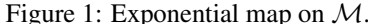
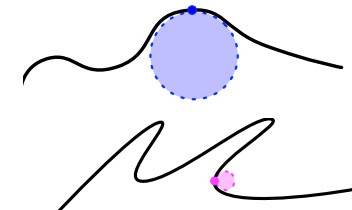

Figure 1: Exponential map on $\mathcal{M}$.     Figure 2: Manifolds with large and small reach.

the sufficiently small $\delta$-ball in the tangent space may vary for each $x \in \mathcal{M}$, we define the injectivity radius of $\mathcal{M}$ as the minimum $\delta$ over all $x \in \mathcal{M}$.

**Definition 2** (Injectivity radius). *For all $x \in \mathcal{M}$, we define the injectivity radius at $x$ to be $inj_{\mathcal{M}}(x) = \sup\{\delta > 0 | \exp_x : B_{T_x\mathcal{M}}(0, \delta) \subseteq T_x\mathcal{M} \to \mathcal{M}$ is a diffeomorphism$\}$. Then the injectivity radius of $\mathcal{M}$ is defined as*

$$inj(\mathcal{M}) = \inf\{inj_{\mathcal{M}}(x) | x \in \mathcal{M}\}.$$

For any $x \in \mathcal{M}$, the exponential map restricted to a ball of radius $inj(\mathcal{M})$ in $T_x\mathcal{M}$ is a well-defined diffeomorphism. Within the injectivity radius, the exponential map is a diffeomorphism between the tangent space and a patch of $\mathcal{M}$, with $\exp^{-1}$ denoting the inverse. Controlling a quantity called reach allows us to lower bound the manifold's injectivity radius.

**Definition 3** (Reach). *The reach $\tau$ of a manifold $\mathcal{M}$ is defined as the quantity (Federer [1959])*

$$\tau = \inf\{r > 0 : \exists x \neq y \in \mathcal{M}, v \in \mathbb{R}^D \text{ such that } r = \|x - v\| = \|y - v\| = \inf_{z \in \mathcal{M}} \|z - v\|\}.$$

Intuitively, if the distance of a point $x$ to $\mathcal{M}$ is smaller than the reach, then there is a unique point in $\mathcal{M}$ that is closest to $x$. However, if the distance between $x$ and $\mathcal{M}$ is larger than the reach, then there will no longer be a unique closest point to $x$ in $\mathcal{M}$. For example, the reach of a sphere is its radius. A manifold with large and small reach is illustrated in Figure 2. The reach gives us control over the injectivity radius $inj(\mathcal{M})$; in particular, we know $inj(\mathcal{M}) \geqslant \pi\tau$ (see Aamari and Levrard [2019] for proof).

## 2.2 Optimal Transport Theory

Let $\mu, \nu$ be absolutely continuous measures on the sets $X, Y \subseteq \mathbb{R}^d$ respectively. We say a function $f : X \to Y$ *transports* $\mu$ onto $\nu$ if $f_\sharp\mu = \nu$. In words, for all measurable sets $A \subseteq Y$ we have

$$\nu(A) = f_\sharp\mu(A) = \mu\left(f^{-1}(A)\right),$$

where $f^{-1}(A)$ is the pre-image of $A$ under $f$.

Optimal transport studies the problem of transporting source measures $\mu$ on $X$ to target measures $\nu$ on $Y$ while minimizing a cost $c : X \times Y \to \mathbb{R}_{\geqslant 0}$. However, the results are largely restricted to transport between measures on the same dimensional Euclidean space. In this paper, we will make use of the main theorem in Caffarelli [1992], in the form presented in Villani [2008].

**Proposition 1.** *Let $c(x, y) = \|x - y\|^2$ in $\mathbb{R}^d \times \mathbb{R}^d$ and let $\Omega_1, \Omega_2$ be nonempty, connected, bounded, open subsets of $\mathbb{R}^d$. Let $f_1, f_2$ be probability densities on $\Omega_1$ and $\Omega_2$ respectively, with $f_1, f_2$ bounded from above and below. Assume further that $\Omega_2$ is convex. Then there exists a unique optimal transport map $T : \Omega_1 \to \Omega_2$ for the associated probability measures $\mu(dx) = f_1(x)\,dx$ and $\nu(dy) = f_2(y)\,dy$, and the cost c. Furthermore, we have that $T \in C^\alpha(\Omega_1)$ for some $\alpha \in (0, 1)$.*

This proposition allows to produce Hölder transport maps which can be further approximated with neural networks with size depending on a given accuracy.

To connect optimal transport and Riemannian manifolds, we first define the *volume measure* on a manifold $\mathcal{M}$ and establish integration on $\mathcal{M}$.

**Definition 4** (Volume measure). *Let $\mathcal{M}$ be a compact $d$-dimensional Riemannian manifold. We define the volume measure $\mu_{\mathcal{M}}$ on $\mathcal{M}$ as the restriction of the $d$-dimensional Hausdorff measure $\mathcal{H}^d$.*

A definition for the restriction of the Hausdorff measure can be found in Federer [1959].

We say that the distribution $Q$ has density $q$ if the Radon-Nikodym derivative of $Q$ with respect to $\mu_{\mathcal{M}}$ is $q$. According to Evans and Gariepy [1992]), for any continuous function $f : \mathcal{M} \to \mathbb{R}$ supported within the image of the ball $B_{T_x\mathcal{M}}(0, \epsilon)$ under the exponential map for $\epsilon < \text{inj}(\mathcal{M})$, we have

$$\int f \, dQ = \int (fq) \, d\mu_{\mathcal{M}} = \int_{B_{T_x\mathcal{M}}(0,\epsilon)} (fq) \circ \exp_x(v) \sqrt{\det g_{ij}^x(v)} \, dv. \tag{5}$$

Here $g_{ij}^x(v) = \langle \partial \exp_x(v)[e_i], \partial \exp_x(v)[e_j] \rangle$ with $(e_1, ..., e_d)$ an orthonormal basis of $T_x\mathcal{M}$.

## 3   Main Results

We will present our main results in this section, including an approximation theory for a large class of distributions on a Riemannian manifold (Theorem 1), and a statistical estimation theory of deep generative networks for distribution learning (Theorem 2).

We make some regularity assumptions on a manifold $\mathcal{M}$ and assume the target data distribution $Q$ is supported on $\mathcal{M}$. The easy-to-sample distribution $\rho$ is taken to be uniform on $(0, 1)^{d+1}$.

**Assumption 1.** *$\mathcal{M}$ is a $d$-dimensional compact Riemannian manifold isometrically embedded in ambient dimension $\mathbb{R}^D$. Via compactness, $\mathcal{M}$ is bounded: there exists $M > 0$ such that $\|x\|_\infty \leqslant M$, $\forall x \in \mathcal{M}$. Further suppose $\mathcal{M}$ has a positive reach $\tau > 0$.*

**Assumption 2.** *$Q$ is supported on $\mathcal{M}$ and has a density $q$ with respect to the volume measure on $\mathcal{M}$. Further we assume boundedness of $q$ i.e. there exists some constants $c, C > 0$ such that $c \leqslant q \leqslant C$.*

To justify the representation power of feedforward ReLU networks for learning the target distribution $Q$, we explicitly construct a neural network generator class, such that a neural network function in this generator class can pushfoward $\rho$ to a good approximation of $Q$.

Consider the following generator class $\mathcal{G}_{\text{NN}}$

$$\mathcal{G}_{\text{NN}}(L, p, \kappa) = \{g = [g_1, ..., g_D] : \mathbb{R}^{d+1} \to \mathbb{R}^D | g_j \text{ in form (3) with at most } L \text{ layers}$$
$$\text{and max width } p, \text{ while } ||W_i||_\infty \leqslant \kappa, ||b_i||_\infty \leqslant \kappa \text{ for all } i \in [L], j \in [D]\},$$

where $\| \cdot \|_\infty$ is the maximum magnitude in a matrix or vector. The width of a neural network is the largest dimension (i.e. number of rows/columns) among the $W_i$'s and $b_i$'s.

**Theorem 1** (Approximation Power of Deep Generative Models). *Suppose $\mathcal{M}$ and $Q$ satisfy Assumptions 1 and 2 respectively. The easy-to-sample distribution $\rho$ is taken to be uniform on $(0, 1)^{d+1}$. Then there exists a constant $0 < \alpha < 1$ (independent of $D$) such that for any $0 < \epsilon < 1$, there exists a $g_\theta \in \mathcal{G}_{\text{NN}}(L, p, \kappa)$ with parameters*

$$L = O\left(\log\left(\frac{1}{\epsilon}\right)\right), \quad p = O\left(D\epsilon^{-\frac{d}{\alpha}}\right), \quad \kappa = M$$

*that satisfies*

$$W_1((g_\theta)_\sharp \rho, Q) < \epsilon.$$

Theorem 1 demonstrates the representation power of deep neural networks for distributions $Q$ on $\mathcal{M}$, which answers Question **Q1**. For a given accuracy $\epsilon$, there exists a neural network $g_\theta$ which pushes the uniform distribution on $(0, 1)^{d+1}$ forward to a good approximation of $Q$ with accuracy $\epsilon$. The network size is exponential in the intrinsic dimension $d$.

We next present a statistical estimation theory to answer Question **Q2**.

**Theorem 2** (Statistical Guarantees of Deep Wasserstein Learning). *Suppose $\mathcal{M}$ and $Q$ satisfy Assumption 1 and 2 respectively. The easy-to-sample distribution $\rho$ is taken to be uniform on $(0, 1)^{d+1}$. Let $n$ be the number of samples of $X_i \sim Q$. Choose any $\delta > 0$. Set $\epsilon = n^{-\frac{1}{d+\delta}}$ in Theorem 1 so that the network class $\mathcal{G}_{\text{NN}}(L, p, \kappa)$ has parameters*

$$L = O\left(\log\left(n^{\frac{1}{d+\delta}}\right)\right), \quad p = O\left(Dn^{\frac{d}{\alpha(d+\delta)}}\right), \quad \kappa = M.$$

*Then the empirical risk minimizer $\hat{g}_n$ given by (4) has rate*

$$\mathbb{E}W_1((\hat{g}_n)_\sharp\rho, Q) \leqslant Cn^{-\frac{1}{d+\delta}},$$

*where $C$ is a constant independent of $n$ and $D$.*

Additionally this result can be easily extended to the noisy case. Suppose we are given $n$ noisy i.i.d. samples $\hat{X}_1, ..., \hat{X}_n$ of the form $\hat{X}_i = X_i + \xi_i$, for $X_i \overset{\text{iid}}{\sim} Q$ and $\xi_i$ distributed according to some noise distribution. The optimization in (4) is performed with the noisy empirical distribution $\hat{Q}_n = \frac{1}{n}\sum_{i=1}^n \delta_{\hat{X}_i}$. Then the minimizer $\hat{g}_n$ satisfies

$$\mathbb{E}W_1((\hat{g}_n)_\sharp\rho, Q) \leqslant Cn^{-\frac{1}{d+\delta}} + 2\sqrt{V_\xi},$$

where $V_\xi = \mathbb{E}\|\xi\|_2^2$ is the variance of the noise distribution.

**Comparison to Related Works.** To justify the practical power of generative networks, low-dimensional data structures are considered in Luise et al. [2020], Schreuder et al. [2021], Block et al. [2021], Chae et al. [2021]. These works consider the generative models in (1). They assume that the high-dimensional data are parametrized by low-dimensional latent parameters. Such assumptions correspond to the manifold model where the manifold is globally homeomorphic to Euclidean space, i.e. the manifold has a single chart.

In Luise et al. [2020], the generative models are assumed to be continuously differentiable up to order $s$. By jointly training of the generator and the latent distributions, they proved that the Sinkhorn divergence between the generated distribution and data distribution converges, depending on data intrinsic dimension. Chae et al. [2021] and Schreuder et al. [2021] assume the special case where the manifold has a single chart. More recently, Block et al. [2021] proposed to estimate the intrinsic dimension of data using the Hölder IPM between some empirical distributions of data. This theory is based on the statistical convergence of the empirical distribution to the data distribution. As an application to GANs, [Block et al., 2021, Theorem 23] gives the statistical error while the approximation error is not studied. In these works, the single chart assumption is very strong while a general manifold can have multiple charts.

Recently, Yang et al. [2022], Huang et al. [2022] showed that GANs can approximate any data distribution (in any dimension) by transforming an absolutely continuous 1D distribution. The analysis in Yang et al. [2022], Huang et al. [2022] can be applied to the general manifold model. Their approach requires the GAN to memorize the empirical data distribution using ReLU networks. Thus it is not clear how the designed generator is capable of generating new samples different from the training data.

In contrast, we explicitly construct an oracle transport map which transforms the low-dimensional easy-to-sample distribution to the data distribution. Our work provides insights about how distributions on a manifold can be approximated by a neural network pushforward of a low-dimensional easy-to-sample distribution without exactly memorizing the data. In comparison, the single-chart assumption in earlier works assumes that an oracle transport naturally exists. Our work is novel in the construction of the oracle transport for a general manifold with multiple charts, and the approximation theory by deep neural networks.

## 4 Proof of Main Results

### 4.1 Proof of Approximation Theory in Theorem 1

To prove Theorem 1, we explicitly construct an oracle transport $g^*$ pushing $\rho$ onto $Q$, i.e. $g^*_\sharp\rho = Q$. Further this oracle will be piecewise $\alpha$-Hölder continuous for some $\alpha \in (0, 1)$.

**Lemma 1.** *Suppose $\mathcal{M}$ and $Q$ satisfy Assumption 1 and 2 respectively. The easy-to-sample distribution $\rho$ is taken to be uniform on $(0, 1)^{d+1}$. Then there exists a function $g^*: (0, 1)^{d+1} \to \mathcal{M}$ such that $Q = g^*_\sharp\rho$ where*

$$g^*(x) = \sum_{j=1}^J \mathbb{1}_{(\pi_{j-1}, \pi_j)}(x_1)g_j^*(x_{2:d+1}) \tag{6}$$

*for some $\alpha$-Hölder ($0 < \alpha < 1$) continuous functions $g_1^*, \ldots, g_J^*$ and some constants $0 = \pi_0 < \pi_1 < \cdots < \pi_J = 1$.*

*Proof.* We construct a transport map $g^* : (0,1)^{d+1} \to \mathcal{M}$ that can be approximated by neural networks. First, we decompose the manifold into overlapping geodesic balls. Next, we pull these local distributions on these balls back to tangent space, which produces $d$-dimensional tangent distributions. Then, we apply optimal theory on these tangent distributions to produce maps between the source distributions on $(0,1)^d$ to the appropriate local (geodesic ball) distributions on the manifold. Finally, we glue together these local maps with indicators functions and a uniform random sample from $(0,1)$. We proceed with the first step of decomposing the manifold.

**Step 1: Overlapping ball decomposition.** Recall that $\mathcal{M}$ is a compact manifold with reach $\tau > 0$. Then the injectivity radius of $\mathcal{M}$ is greater than or equal to $\pi\tau$ (Aamari et al. [2019]). Set $r = \frac{\pi\tau}{2}$. For each $c \in \mathcal{M}$, define an open set $U_c = \exp_c(B_{T_c\mathcal{M}}(0,r)) \subseteq \mathcal{M}$. Since the collection $\{U_c : x \in \mathcal{M}\}$ forms an open cover of $\mathcal{M}$ (in $\mathbb{R}^D$), by the compactness of $\mathcal{M}$ we can extract a finite subcover which we denote as $\{U_{c_j}\}_{j=1}^J$. For convenience, we will write $U_j = U_{c_j}$.

**Step 2: Defining local lower-dimensional distributions.** On each $U_j$, we define a local distribution $Q_j$ with density $q_j$ via

$$q_j(x) = \frac{q(x)}{Q(U_j)} \mathbb{1}_{U_j}(x).$$

Set $K(x) = \sum_{j=1}^J \mathbb{1}_{U_j}(x)$ as the number of balls $U_j$ containing $x$. Note $1 \leqslant K(x) \leqslant J$ for all $x \in \mathcal{M}$. Now define the distribution $\overline{Q}_j$ with density $\overline{q}_j$ given by

$$\overline{q}_j(x) = \frac{\frac{1}{K(x)} q_j(x) \mathbb{1}_{U_j}(x)}{\int_{U_j} \frac{1}{K(x)} q_j(x) d\mathcal{H}}.$$

Write $K_j = \int_{U_j} \frac{1}{K(x)} q_j(x) d\mathcal{H}$ as the normalizing constant. Define $\tilde{q}_j(v) = (\overline{q}_j \circ \exp_{c_j})(v)\sqrt{\det g_{kl}^{c_j}(v)}$ where $g_{kl}^{c_j}$ is the Riemannian at $c_j$. This quantity can be thought of as the Jacobian of the exponential map, denoted by $|J_{\exp_{c_j}}(v)|$ in the following step. Then $\tilde{q}_j$ is a density on $\tilde{U}_j = \exp_{c_j}^{-1}(U_j)$, which is a ball of radius $\frac{\pi\tau}{2}$ since

$$1 = \int_{U_j} \overline{q}_j(x) d\mathcal{H} = \int_{\tilde{U}_j} \sqrt{\det g_{kl}^{c_j}(v)} \overline{q}_j(\exp_{c_j}(v)) dv = \int_{\tilde{U}_j} \tilde{q}_j(v) dv$$

Let $\tilde{Q}_j$ be the distribution in $\mathbb{R}^d$ with density $\tilde{q}_j$. By construction, we can write

$$\overline{Q}_j = (\exp_{c_j})_\sharp \tilde{Q}_j. \tag{7}$$

**Step 3: Constructing the local transport.** We have that $\exp_{c_j}^{-1}$ is bi-Lipschitz on $U_j$ and hence its Jacobian is upper bounded. Since $|J_{\exp_{c_j}}(v)| = \frac{1}{|J_{\exp_{c_j}^{-1}}(x)|}$, we know that $|J_{\exp_{c_j}}|$ lower bounded. Since $q_j$ is lower bounded (away from 0), this means $\tilde{q}_j$ is also lower bounded. Now the distribution $\tilde{\rho}_j$ supported on $\tilde{U}_j = B(0, \frac{\tau\pi}{2})$ fulfills the requirements for our optimal transport result: (1) Its density $\tilde{p}_j$ is lower and upper bounded; (2) The support $B(0, \frac{\tau\pi}{2})$ is convex. Taking our cost to be $c(x,y) = \frac{1}{2}\|x-y\|^2$ (i.e. squared Euclidean distance), via Proposition 1 we can find an optimal transport map $T_j$ such that

$$(T_j)_\sharp \rho_d = \tilde{Q}_j \tag{8}$$

where $\rho_d$ is uniformly distributed on $(0,1)^d$. Furthermore, $T_j \in C^{\alpha_j}$ for some $\alpha_j \in (0,1)$. Then we can construct a local transport onto $U_j$ via

$$g_j^* = \exp_{c_j} \circ T_j \tag{9}$$

which pushes $\rho_d$ forward to $\overline{Q}_j$. Since $g_j^*$ is a composition of a Lipschitz map with an $\alpha_j$ Hölder continuous maps, it is hence $\alpha_j$ Hölder continuous.

**Step 4: Assembling the global transport.** It remains to patch together the local distributions $\overline{Q}_j$ to form $Q$. Define $\eta_j = K_j Q(U_j)$. Notice

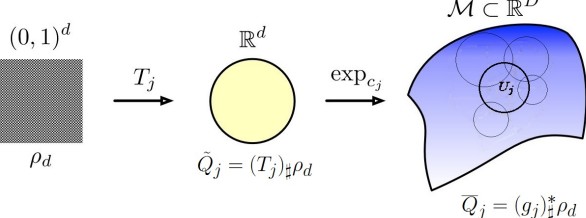

Figure 3: Local transport $g_j^*$ in (9) mapping $\rho_d$ on $(0,1)^d$ to a local distribution $\overline{Q}_j$ supported on $U_j$.

$$\sum_{j=1}^{J} \eta_j \overline{q}_j(x) = \sum_{j=1}^{J} K_j Q(U_j) \frac{\frac{1}{K(x)} q_j(x) \mathbb{1}_{U_j}(x)}{K_j} = \sum_{j=1}^{J} Q(U_j) \frac{\frac{1}{K(x)} q(x) \mathbb{1}_{U_j}(x)}{Q(U_j)}$$

$$= \sum_{j=1}^{J} \frac{1}{K(x)} q(x) \mathbb{1}_{U_j}(x) = q(x) \frac{1}{K(x)} \sum_{j=1}^{J} \mathbb{1}_{U_j}(x) = q(x).$$

Hence it must be that $\sum_{j=1}^{J} \eta_j = 1$. Set $\alpha = \min_{j \in [J]} \alpha_j$. We can now define the oracle $g^*$. Let $x \in (0,1)^{d+1}$. Write

$$g^*(x) = \sum_{j=1}^{J} \mathbb{1}_{(\pi_{j-1}, \pi_j)}(x_1) g_j^*(x_{2:d+1}), \tag{10}$$

where $x_1$ is the first coordinate and $x_{2:d+1}$ are the remaining coordinates with $\pi_j = \sum_{i=1}^{j-1} \eta_i$. Let $Z \sim \rho$. Then $g(Z) \sim Q$. We see this as follows. For $A \subseteq \mathcal{M}$ we can compute

$$\mathbb{P}(g^*(Z) \in A) = \sum_{j=1}^{J} \mathbb{P}(\pi_{j-1} < Z_1 < \pi_j) \mathbb{P}(g_j^*(Z_{2:d+1}) \in A \cap U_j) = \sum_{j=1}^{J} \eta_j \overline{Q}_j(A \cap U_j)$$

$$= \sum_{j=1}^{J} \eta_j \int_A \overline{q}_j(x) d\mathcal{H} = \int_A \sum_{j=1}^{J} \eta_j \overline{q}_j(x) d\mathcal{H} = \int_A q(x) d\mathcal{H} = Q(A)$$

which completes the proof. $\qquad\square$

We have found an oracle $g^*$ which is piecewise Hölder continuous such that $g_\sharp^* \rho = Q$. We can design a neural network $g_\theta$ to approximate this oracle $g^*$. Now in order to minimize $W_1((g_\theta)_\sharp \rho, Q) = W_1((g_\theta)_\sharp \rho, g_\sharp^* \rho)$, we show it suffices to have $g_\theta$ approximate $g^*$ in $L^1(\rho)$.

**Lemma 2.** *Let $\mu$ be an absolutely continuous probability distribution on a set $Z \subseteq \mathbb{R}^d$, and let $f, g : Z \to \mathbb{R}^m$ be transport maps. Then*

$$W_1(f_\sharp \mu, g_\sharp \mu) \leqslant C \|f - g\|_{L^1(\mu)}$$

*for some $C > 0$.*

The proof can be found in Section B.1 in the appendix. We now prove Theorem 1.

*Proof of Theorem 1.* By Lemma 1, there exists a transformation $g^*(x) = \sum_{j=1}^{J} \mathbb{1}_{(\pi_{j-1}, \pi_j)}(x_1) g_j^*(x_{2:d+1})$ such that $g_\sharp^* \rho = Q$. By Lemma 2, it suffices to approximate $g^*$ with a neural network $g_\theta \in \mathcal{G}_{\mathrm{NN}}(L, p, \kappa)$ in $L^1$ norm, with a given accuracy $\epsilon > 0$. Let $(g^*)^{(i)}$ denote the $i$th component of the vector valued function $g^*$. Then it suffices to approximate

$$(g^*)^{(i)}(x) = \sum_{j=1}^{J} \mathbb{1}_{(\pi_{j-1}, \pi_j)}(x_1) (g_j^*)^{(i)}(x_{2:d+1})$$

for each $1 \leqslant i \leqslant D$, where $(g_j^*)^{(i)}$ denotes the $i$th component of the function $g_j^*$. We construct the approximation of $(g^*)^{(i)}$ by the function

$$(g_\theta)^{(i)}(x) = \sum_{j=1}^{J} \tilde{\times}^{\delta_2} \left( \tilde{\mathbb{1}}^{\delta_1}_{(\pi_{j-1}, \pi_j)}(x_1), (g_{j,\theta}^{\delta_3})^{(i)}(x_{2:d+1}) \right), \tag{11}$$

where $\tilde{\times}^{\delta_2}$ is a ReLU network approximation to the multiplication operation with $\delta_2$ accuracy, $\tilde{\mathbb{1}}^{\delta_1}_{(\pi_{j-1}, \pi_j)}$ is a ReLU network approximation to the indicator function with $\delta_1$ accuracy, and $(g_{j,\theta}^{\delta_3})^{(i)}$ is a ReLU network approximation to $(g_j^*)^{(i)}$ with $\delta_3$ accuracy. We construct these using the approximation theory outlined in Appendix A.

First, we obtain $\tilde{\mathbb{1}}^{\delta_1}_{(\pi_{j-1}, \pi_j)}$ via an application of Lemma 9. Next, we obtain $\tilde{\times}^{\delta_2}$ from an application of Lemma 7. Finally, we discuss $g_{j,\theta}^{\delta_3}$. Let $j \in [J]$. To approximate the Hölder function $g_j^*$, we use the following Lemma 3 that is proved in Appendix A. Similar approximation results can be found in Shen et al. [2022] and Ohn and Kim [2019] as well. In Lemma 3, our approximation error is in $L^1$ norm and all weight parameters are upper bounded by a constant. In comparison, the error in Ohn and Kim [2019] is in $L^\infty$ norm and the weight parameter increases as $\epsilon$ decreases.

**Lemma 3.** *Fix $M \geqslant 2$. Suppose $f \in C^\alpha([0,1]^d)$, $\alpha \in (0,1]$, with $\|f\|_{L^\infty} < M$. Let $0 < \epsilon < 1$. Then there exists a function $\Phi$ implementable by a ReLU network such that*

$$\|f - \Phi\|_{L^1} < \epsilon.$$

*The ReLU network has depth at most $c_1 \log \left( \frac{1}{\epsilon} \right)$, width at most $c_2 \epsilon^{-\frac{d}{\alpha}}$, and weights bounded by $M$ (where $c_1$ and $c_2$ are constants independent of $\epsilon$).*

We can apply Lemma 3 to $(g_j^*)^{(i)}$ for all $1 \leqslant j \leqslant J$ and $1 \leqslant i \leqslant D$, since they are all elements of $C^\alpha(0,1)^d$ and elements of $C^\alpha(0,1)^d$ can be extended to $C^\alpha[0,1]^d$. Thus there exists a neural network $(g_{j,\theta}^{\delta_3})^{(i)} \in \mathcal{G}_{NN}(L, p, \kappa)$ with parameters given as above such that

$$\|(g_j^*)^{(i)} - (g_{j,\theta}^{\delta_3})^{(i)}\|_{L^1} < \delta_3.$$

The goal is now to show the $L^1$ distance between $g_\theta$ (as defined in (11)) and $g^*$ is small. We have

$$\|g^* - g_\theta\|_{L^1} \leqslant DJ(M\delta_1 + \delta_2 + \delta_3)$$

where $\delta_1$ is the neural network approximation error of indicator functions in $L^1$, $\delta_2$ is the approximation error of multiplication, and $\delta_3$ is the approximation error of our $\alpha$-Hölder local transport maps. We carefully argue in Appendix A.4 that each of these components can be approximated with the appropriately sized network. To complete the proof we conclude $g_\theta$ can be exactly represented by a neural network in $\mathcal{G}_{NN}(L, p, \kappa)$ with parameters

$$L = O\left( \log \left( \frac{1}{\epsilon} \right) \right), \quad p = O\left( D\epsilon^{-\frac{d}{\alpha}} \right), \quad \kappa = M.$$

$\square$

## 4.2 Proof of Statistical Estimation Theory in Theorem 2

The proof of Theorem 2 is facilitated by the common bias-variance inequality, presented here as a lemma.

**Lemma 4.** *Under the same assumptions of Theorem 2, we have*

$$\mathbb{E} W_1((\hat{g}_n)_\sharp \rho, Q) \leqslant \inf_{g_\theta \in \mathcal{G}_{NN}} W_1((g_\theta)_\sharp \rho, Q) + 2\mathbb{E} W_1(Q_n, Q) \tag{12}$$

*where $Q_n$ is the empirical distribution.*

*Proof.* We compute recalling the definition of $\hat{g}_n$ as the empirical risk minimizer.

$$\begin{aligned}
\mathbb{E}W_1((\hat{g}_n)_\sharp\rho, Q) &\leqslant \mathbb{E}W_1((\hat{g}_n)_\sharp\rho, Q_n) + \mathbb{E}W_1(Q_n, Q) \\
&= \mathbb{E}\inf_{g_\theta \in \mathcal{G}_{\mathrm{NN}}} W_1((g_\theta)_\sharp\rho, Q_n) + \mathbb{E}W_1(Q_n, Q) \\
&\leqslant \mathbb{E}\inf_{g_\theta \in \mathcal{G}_{\mathrm{NN}}} W_1((g_\theta)_\sharp\rho, Q) + 2\mathbb{E}W_1(Q_n, Q)
\end{aligned}$$

where we recall $W_1((\hat{g}_n)_\sharp\rho, Q_n) = \inf_{g_\theta \in \mathcal{G}_{\mathrm{NN}}} W_1((g_\theta)_\sharp\rho, Q_n)$ from (4).

$\square$

The bias term can be controlled via Theorem 1. To control convergence of the empirical distribution $Q_n$ to $Q$ we leverage the existing theory [Weed and Bach, 2019] to obtain the following lemma.

**Lemma 5.** *Under the same assumptions of Theorem 2, for all $\delta > 0$, $\exists C_\delta > 0$ such that*

$$\mathbb{E}\left[W_1(Q, Q_n)\right] \leqslant C_\delta n^{-\frac{1}{d+\delta}}. \tag{13}$$

This follows directly from Theorem 1 from [Weed and Bach, 2019]. We attach a full proof in Section B.2 of the appendix. Finally, we prove our statistical estimation result in Theorem 2.

*Proof of Theorem 2.* Choose $\delta > 0$. Recall from Lemma 4 we have

$$W_1((\hat{g}_n)_\sharp\rho, Q) \leqslant \mathbb{E}\inf_{g_\theta \in \mathcal{G}_{\mathrm{NN}}} W_1((g_\theta)_\sharp\rho, Q) + 2\mathbb{E}W_1(Q_n, Q)$$

The first term is the approximation error which can be controlled within an arbitrarily small accuracy $\epsilon$. Theorem 1 shows the existence of a neural network function $g_\theta$ with $O\left(\log\left(\frac{1}{\epsilon}\right)\right)$ layers and $O(D\epsilon^{-d/\alpha}\log(\frac{1}{\epsilon}))$ neurons such that $W_1((g_\theta)_\sharp\rho, Q) \leqslant \epsilon$ for any $\epsilon > 0$. We choose $\epsilon = n^{-\frac{1}{d+\delta}}$ to optimally balance the approximation error and the statistical error. The second term is the statistical error for which we recall from Lemma 5 that $\mathbb{E}\left[W_1(\hat{Q}_n, Q)\right] \leqslant C_\delta n^{-\frac{1}{d+\delta}}$ for some constant $C_\delta$.

Thus we have

$$\mathbb{E}W_1((\hat{g}_n)_\sharp\rho, Q) \leqslant n^{-\frac{1}{d+\delta}} + 2C_\delta n^{-\frac{1}{d+\delta}} = Cn^{-\frac{1}{d+\delta}}$$

by setting $C = 1 + 2C_\delta$. This concludes the proof. $\square$

We remark the above proof proceeds similarly in the noisy case, which is presented in Section B.3 of the appendix.

## 5 Conclusion

We have established approximation and statistical estimation theories of deep generative models for estimating distributions on a low-dimensional manifold. The statistical convergence rate in this paper depends on the intrinsic dimension of data. In light of the manifold hypothesis, which suggests many natural datasets lie on low dimensional manifolds, our theory rigorously explains why deep generative models defy existing theoretical sample complexity estimates and the curse of dimensionality. In fact, deep generative models are able to learn low-dimensional geometric structures of data, and allow for highly efficient sample complexity independent of the ambient dimension. Meanwhile the size of the required network scales exponentially with the intrinsic dimension.

Our theory imposes very little assumption on the target density $Q$, requiring only that it admit a density $q$ with respect to the volume measure and that $q$ is upper and lower bounded. In particular we make no smoothness assumptions on $q$. This is practical, as we do not expect existing natural datasets to exhibit high degrees of smoothness.

In this work, we assume access to computation of the $W_1$ distance. However during GAN training a discriminator is trained for this purpose. It would be of interest for future work to investigate the low-dimensional role of such discriminator networks which approximate the $W_1$ distance in practice.

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
