# A Deep ReLU Approximation of Hölder functions

In this section, $\log$ denotes the base 2 logarithm by default. $\times_{i=1}^{d}$ denotes the Cartesian product of $d$ sets. The goal is to determine the approximation rate of deep ReLU networks for Hölder continuous functions. Let $f \in C^{\alpha}([0,1]^d)$ with Hölder norm $\|f\|_{C^{\alpha}}$ where $\alpha \in (0,1)$. We first approximate $f$ by a piecewise constant function $f^n$ in Section A.1, and then approximate $f^n$ by a deep ReLU Network $\Phi$ in Section A.3.

## A.1 Piecewise constant approximation

Let $[n]^d = \{(a_1, a_2, \ldots, a_d) : a_i \in \mathbb{N}, 1 \leqslant a_i \leqslant n\}$. Given any $n \in \mathbb{N}$, we cover $[0,1]^d$ by $n^d$ non-overlapping open cubes. For any $(k_1, \ldots, k_d) = \vec{k} \in [n]^d$, we define

$$Q_{\vec{k}} = \mathop{\times}_{i=1}^{d} \left( \frac{k_i - 1}{n}, \frac{k_i}{n} \right). \tag{14}$$

**Lemma 6.** *Let $f \in C^{\alpha}([0,1]^d)$ with Hölder norm $\|f\|_{C^{\alpha}}$. For any $n \in \mathbb{N}$, define*

$$f^n(x) = \sum_{k \in [n]^d} \left( n^d \int_{Q_{\vec{k}}} f(y)\, dy \right) \mathbb{1}_{Q_{\vec{k}}}(x).$$

*Then*

$$\|f - f^n\|_{L^1} < \|f\|_{C^{\alpha}} \frac{d^{\alpha/2}}{n^{\alpha}}.$$

*Proof.* We estimate

$$\|f - f^n\|_{L^1} = \int |f(x) - f^n(x)|\, dx$$

$$= \int \left| \sum_{k \in [n]^d} \left( n^d \int_{Q_{\vec{k}}} f(x)\, dy \right) \mathbb{1}_{Q_{\vec{k}}}(x) - \sum_{k \in [n]^d} \left( n^d \int_{Q_{\vec{k}}} f(y)\, dy \right) \mathbb{1}_{Q_{\vec{k}}}(x) \right| dx$$

$$= n^d \int \left| \sum_{k \in [n]^d} \left( \int_{Q_{\vec{k}}} (f(x) - f(y))\, dy \right) \mathbb{1}_{Q_{\vec{k}}}(x) \right| dx$$

$$\leqslant n^d \sum_{k \in [n]^d} \int \mathbb{1}_{Q_{\vec{k}}}(x) \int_{Q_{\vec{k}}} |f(x) - f(y)|\, dy\, dx$$

$$= n^d \sum_{k \in [n]^d} \int_{Q_{\vec{k}}} \int_{Q_{\vec{k}}} |f(x) - f(y)|\, dy\, dx$$

$$\leqslant n^d \sum_{k \in [n]^d} \int_{Q_{\vec{k}}} \int_{Q_{\vec{k}}} \|f\|_{C^{\alpha}} \frac{d^{\alpha/2}}{n^{\alpha}}\, dy\, dx$$

$$= \|f\|_{C^{\alpha}} \frac{d^{\alpha/2}}{n^{\alpha}} n^d \sum_{k \in [n]^d} \frac{1}{n^{2d}} = \|f\|_{C^{\alpha}} \frac{d^{\alpha/2}}{n^{\alpha}}.$$

where we use crucially use the fact that $\displaystyle\sup_{x,y \in Q_{\vec{k}}} |f(x) - f(y)| \leqslant \|f\|_{C^{\alpha}} \sup_{x,y \in Q_{\vec{k}}} |x-y|^{\alpha} = \|f\|_{C^{\alpha}} \frac{d^{\alpha/2}}{n^{\alpha}}$. $\square$

## A.2 Neural network approximation

We start with the well-known result originally stated in Yarotsky [2017].

**Lemma 7.** *Let $A > 0$. For any $\epsilon \in (0, A^2)$, there is a ReLU network which implements a function $\tilde{\times} : \mathbb{R}^2 \to \mathbb{R}$ such that*

$$\sup_{|x| \leqslant A, |y| \leqslant A} \left| \tilde{\times}(x, y) - xy \right| = \epsilon.$$

*This network has depth at most $c \log\left(\frac{A^2}{\epsilon}\right)$, width at most 8, and weights bounded by $A$ (where $c$ is an absolute constant).*

*Proof.* The result follows from a careful reading of the proof in Appendix A.2 in Chen et al. [2019]. □

The network given by Lemma 7 approximates the multiplication of two numbers. We seek an approximation of the multiplication of $d$ numbers, and this is achieved by composing $\tilde{\times}$ with itself.

**Lemma 8.** *Fix $d \in \mathbb{N}$ and let $M > 0$. For any $\epsilon \in (0, M^2)$, there is a ReLU network which implements a function $\tilde{\times}_d : \mathbb{R}^d \to \mathbb{R}$ such that*

$$\sup_{|x_1|, \ldots, |x_d| \leqslant M} \left| \tilde{\times}(x_1, \ldots, x_d) - x_1 \cdots x_d \right| < \epsilon.$$

*This network has depth at most $c_1 \log\left(\frac{d^3 M^d}{\epsilon}\right) + c_2$, width at most $8d$, and weights bounded by $2M$ (where $c_1$ and $c_2$ are absolute constants).*

*Proof.* Our idea is to realize the multiplication in a binary tree structure, illustrated in Figure 4. We first assume that $2^{k-1} < d \leqslant 2^k$ for some integer $k$, and let $\delta = \frac{\epsilon}{4^{k-1} M^{2^k - 2}}$. We first handle the case that $d = 2^k$. We will construct a family of $k$ functions $\left\{ \tilde{\times}_{2^i} : \mathbb{R}^{2^i} \to \mathbb{R} \right\}_{i=1}^{k}$ iteratively. We will show that for all $1 \leqslant i \leqslant k$, the function $\tilde{\times}_{2^i}$ implements $2^i$-ary multiplication with error at most $4^{i-1} M^{2^i - 2} \delta$ (when $|x_i| < M$), at most $c \log\left(\frac{M^{2^{i+1} - 2}}{\delta^i}\right)$ layers, width at most $4 \cdot 2^i$, and weights bounded by $M$.

For $i = 1$, we define $\tilde{\times}_2$ to be the function defined in Lemma 7 with the parameters $\epsilon = \delta$ and $A = M$. Then $\tilde{\times}_2$ has maximum error $\delta = 4^{1-1} M^{2^1 - 2} \delta$ and is implementable by a ReLU network with at most $c \log\left(\frac{A^2}{\epsilon}\right) = c \log\left(\frac{M^2}{\delta}\right) = c \log\left(\frac{M^{2^{1+1} - 2}}{\delta^1}\right)$ layers, width $8 = 4 \cdot 2^1$, and weights bounded by $M$, all as desired.

Now suppose the claim has been proven for $\tilde{\times}_{2^i}$. Let $\tilde{\times}$ be the function defined in Lemma 7 with the parameters $\epsilon = \delta$ and $A = M^{2^i}$. Then $\tilde{\times}$ has $c \log\left(\frac{A^2}{\epsilon}\right) = c \log\left(\frac{M^{2^{i+1}}}{\delta}\right)$ layers, width 8, and weights bounded by $M$. We define

$$\tilde{\times}_{2^{i+1}}(x_1, \ldots, x_{2^{i+1}}) = \tilde{\times}\left(\tilde{\times}_{2^i}(x_1, \ldots, x_{2^i}), \tilde{\times}_{2^i}(x_{2^i+1}, \ldots, x_{2^{i+1}})\right).$$

Then $\tilde{\times}_{2^{i+1}}$ has depth $c \log\left(\frac{M^{2^{i+1} - 2}}{\delta^i}\right) + c \log\left(\frac{M^{2^{i+1}}}{\delta}\right) = c \log\left(\frac{M^{2^{i+2} - 2}}{\delta^{i+1}}\right)$, width $4 \cdot 2^i + 4 \cdot 2^i = 4 \cdot 2^{i+1}$, and weights bounded by $M$. It remains to compute the following error bound:

$$
\begin{aligned}
&\left| \tilde{\times}\left(\tilde{\times}_{2^i}(x_1, \ldots, x_{2^i}), \tilde{\times}_{2^i}(x_{2^i+1}, \ldots, x_{2^{i+1}})\right) - x_1 \cdots x_{2^{i+1}} \right| \\
&\leqslant \left| \tilde{\times}\left(\tilde{\times}_{2^i}(x_1, \ldots, x_{2^i}), \tilde{\times}_{2^i}(x_{2^i+1}, \ldots, x_{2^{i+1}})\right) - \tilde{\times}_{2^i}(x_1, \ldots, x_{2^i}) \cdot \tilde{\times}_{2^i}(x_{2^i+1}, \ldots, x_{2^{i+1}}) \right| \\
&\quad + \left| \tilde{\times}_{2^i}(x_1, \ldots, x_{2^i}) \cdot \tilde{\times}_{2^i}(x_{2^i+1}, \ldots, x_{2^{i+1}}) - \tilde{\times}_{2^i}(x_1, \ldots, x_{2^i}) \cdot x_{2^i+1} \cdots x_{2^{i+1}} \right| \\
&\quad + \left| \tilde{\times}_{2^i}(x_1, \ldots, x_{2^i}) \cdot x_{2^i+1} \cdots x_{2^{i+1}} - x_1 \cdots x_{2^{i+1}} \right| \\
&\leqslant \delta + \left| \tilde{\times}_{2^i}(x_1, \ldots, x_{2^i}) \right| \cdot \left| \tilde{\times}_{2^i}(x_{2^i+1}, \ldots, x_{2^{i+1}}) - x_{2^i+1} \cdots x_{2^{i+1}} \right| \\
&\quad + \left| x_{2^i+1} \ldots x_{2^{i+1}} \right| \cdot \left| \tilde{\times}_{2^i}(x_1, \ldots, x_{2^i}) - x_1 \cdots x_{2^i} \right| \\
&\leqslant \delta + M^{2^i}(4^{i-1} M^{2^i - 2} \delta) + M^{2^i}(4^{i-1} M^{2^i - 2} \delta)\delta \\
&= (1 + 24^{i-1} M^{2^{i+1} - 2})\delta \\
&< 4^i M^{2^{i+1} - 2} \delta.
\end{aligned}
$$

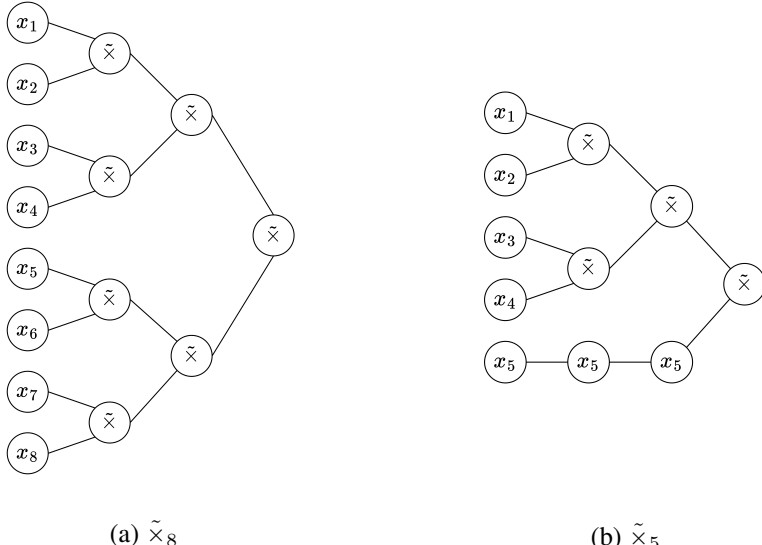

(a) $\tilde{\times}_8$                (b) $\tilde{\times}_5$

Figure 4: Network diagrams for ReLU networks approximating multiplication in Lemma 8.

From this, we have constructed a function $\tilde{\times}_{2^k}$ that approximates multiplication (of values $< M$) with error at most $4^{k-1}M^{2^k-2}\delta = \epsilon$ that has depth

$$
\begin{aligned}
c\log\left(\frac{M^{2^{k+1}-2}}{\delta^k}\right) &= c\log\left(\frac{M^{2^k-2}M^{2^k}(4^{k-1})^k(M^{2^k-2})^k}{\epsilon^k}\right) \\
&= c\log\left(M^{2^k-2}\right) + c\log\left(\frac{(4^{k-1})^k(M^{2^k})^k}{\epsilon^k}\right) \\
&= c\log\left(M^{2^k-2}\right) + ck\log\left(\frac{4^{k-1}M^{2^k}}{\epsilon}\right) \\
&< c(1+k)\log\left(\frac{4^{k-1}M^{2^k}}{\epsilon}\right) \\
&< c_1 k\log\left(\frac{4^{k-1}M^{2^k}}{\epsilon}\right),
\end{aligned}
$$

For some absolute constant $c_1$. Now since $k = \log(d)$, we have that $M^{2^k} = M^d$ and $4^{k-1} < 4^k = d^2$, so the ReLU network has depth at most $c\log(d)\log\left(\frac{d^2 M^d}{\epsilon}\right)$ where $c$ is an absolute constant (the same constant as in Lemma 7). The width of $\tilde{\times}_{2^k}$ is $4 \cdot 2^k = 4d$, and the weights are bounded by $M$.

Figure 4(a) shows a neural network diagram for the ReLU network implementing $\tilde{\times}_8$, which has the structure of a full binary tree. In order to handle numbers that are not powers of two, we use an architecture similar to the diagram in Figure 4 (b) which depicts the ReLU network implementing $\tilde{\times}_5$.

Formally, suppose we have $2^{i-1} < d \leqslant 2^i$ for some $i \in \mathbb{N}$. Then consider the network $\tilde{\times}_{2^i}$ defined as before, but we remove the last $2^i - d$ input neurons, and replace them with $1$ everywhere they appear. Note that this can be achieved by adjusting the bias of each neuron appropriately. For example, any neuron can be turned into a constant $1$ by making the weight vector $0$ and making the bias equal to $1$. This procedure will not affect the number of layers, it will not increase the width, and the parameters are bounded by $M + 1 < 2M$. Noting that $2^i < 2d$, we see that the ReLU network has width at most $4 \cdot 2^i < 4 \cdot 2d = 8d$. Finally, the depth is at most (for $c_1$ and $c_2$ absolute constants)

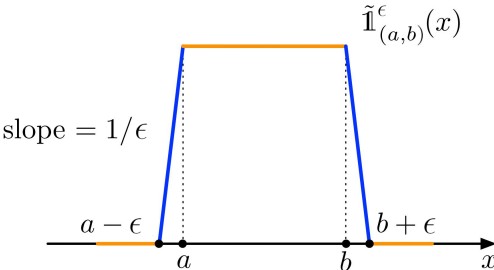

Figure 5: Plot of $\tilde{\mathbb{1}}^{\epsilon}_{(a,b)}$.

$$c\log(2^i)\log\left(\frac{(2^i)^2 M^d}{\epsilon}\right) < c\log(2d)\log\left(\frac{(2d)^2 M^d}{\epsilon}\right)$$

$$= c(\log(2)+\log(d))\left(\log(4)+\log\left(\frac{d^2 M^d}{\epsilon}\right)\right)$$

$$< c\left(\log(8)+3\log(d)\log\left(\frac{d^2 M^d}{\epsilon}\right)\right)$$

$$= c_1\log\left(\frac{d^3 M^d}{\epsilon}\right)+c_2.$$

$\square$

Next we approximate the indicator functions of intervals (which we denote by $\mathbb{1}_{(a,b)}$).

**Lemma 9.** *Fix $M > 1$. Let $[a, b] \subseteq [-M, M]$ and $\epsilon < \frac{1}{2}(b-a)$. Then there is a ReLU network which implements a function $\tilde{\mathbb{1}}^{\epsilon}_{(a,b)}$ such that*

$$\left\|\tilde{\mathbb{1}}^{\epsilon}_{(a,b)} - \mathbb{1}_{(a,b)}\right\|_{L^1} = \epsilon.$$

*This network has depth at most $c\log\left(\frac{1}{\epsilon}\right)$, width equal to 4, and weights bounded by $M$ (where $c$ is a constant depending only on $M$).*

*Proof.* We define the ReLU network function

$$\tilde{\mathbb{1}}^{\epsilon}_{(a,b)}(x) = \frac{1}{\epsilon}(\sigma(x-(a-\epsilon)) - \sigma(x-a) - \sigma(x-b) + \sigma(x-(b+\epsilon))),$$

where $\sigma$ is the ReLU activation function. Figure 5 is a plot of $\tilde{\mathbb{1}}^{\epsilon}_{(a,b)}$. Then it is clear that

$$\left\|\tilde{\mathbb{1}}^{\epsilon}_{(a,b)} - \mathbb{1}_{(a,b)}\right\|_{L^1} = \epsilon.$$

Note that we can express $\frac{1}{\epsilon}$ as

$$\frac{1}{\epsilon} = M^{\lceil\log_M\left(\frac{1}{\epsilon}\right)\rceil-1} \cdot \frac{\frac{1}{\epsilon}}{M^{\lceil\log_M\left(\frac{1}{\epsilon}\right)\rceil-1}} = \underbrace{M \times M \times \cdots \times M}_{\lceil\log_M\left(\frac{1}{\epsilon}\right)\rceil-1 \text{ times}} \times R,$$

where $0 < R \leqslant M$. This is a product of $\lceil\log_M\left(\frac{1}{\epsilon}\right)\rceil$ numbers that are all bounded by $M$. Then note that

$$\tilde{\mathbb{1}}^{\epsilon}_{(a,b)} = M \times \cdots \times M \times R(\sigma(x-(a-\epsilon)) - \sigma(x-a) - \sigma(x-b) + \sigma(x-(b+\epsilon)))$$

can be implemented by ReLU network with $1 + \lceil\log_M\left(\frac{1}{\epsilon}\right)\rceil$ layers. The first layer has 4 neurons, and the second layer has one neuron, and they together compute $R(\sigma(x-(a-\epsilon)) - \sigma(x-a) - \sigma(x-$

$b) + \sigma(x - (b + \epsilon)))$. Then the next $\left\lceil \log_M \left( \frac{1}{\epsilon} \right) \right\rceil - 1$ each multiply this value by $M$ (since all values at this point are positive, the ReLU activation does nothing at each layer). Thus the ReLU network has width 4 (though only the first layer has more than one neuron) and weights bounded by $M$. $\quad\square$

We combine Lemma 8 and Lemma 9 to obtain an approximation to the indicator function of $d$-dimensional cube.

**Lemma 10.** *Fix $M > 1$. Let $Q = \times_{k=1}^d (a_k, b_k) \subseteq [-M, M]^d$ be a bounded $d$-dimensional cube (i.e. $b_1 - a_1 = b_k - a_k$ for all $k \in [d]$), and suppose $\epsilon < \min \left( \frac{b_1 - a_1}{2}, 1 \right)$. Then there exists a function $\phi : [-M, M]^d \to \mathbb{R}$ implementable by a ReLU network such that*

$$\int_{[-M,M]^d} |\mathbb{1}_Q(x) - \phi(x)| \, dx < \epsilon.$$

*The network has depth at most $c_1 \log \left( \frac{d^2 4^d}{\epsilon} \right) + c_2$, width at most $4d$, and weights bounded by $\max\{M, 2\}$ (where $c_1$ and $c_2$ are constants only depending on $M$).*

*Proof.* Denote by $\tilde{\mathbb{1}}^\delta_{(a_i, b_i)}$ the approximation to $\mathbb{1}_{(a_i, b_i)}$ obtained from Lemma 9 with $\delta = \frac{\epsilon}{2}$. Let $\eta = \frac{\epsilon}{2^{d+1} M^d}$, and denote by $\tilde{\times}_d$ the approximation of the multiplication of $d$ factors obtained from Lemma 8 with parameters $M = 1$ and error $\eta$ (which is denoted as $\epsilon$ in the lemma statement). Then we define $\phi$ by

$$\phi(x_1, \ldots, x_d) = \tilde{\times}_d \left( \tilde{\mathbb{1}}^\delta_{(a_1, b_1)}(x_1), \ldots, \tilde{\mathbb{1}}^\delta_{(a_d, b_d)}(x_d) \right).$$

We compute

$$\int_{[-M,M]^d} \left| \mathbb{1}_{\times_{i=1}^d [a_i, b_i]}(x) - \phi(x) \right| \, dx$$

$$= \int_{[-M,M]^d} \left| \prod_{i=1}^d \mathbb{1}_{(a_i, b_i)}(x_i) - \tilde{\times}_d \left( \tilde{\mathbb{1}}^\delta_{(a_1, b_1)}(x_1), \ldots, \tilde{\mathbb{1}}^\delta_{(a_d, b_d)}(x_d) \right) \right| \, dx$$

$$\leq \int_{[-M,M]^d} \left| \prod_{i=1}^d \mathbb{1}_{(a_i, b_i)}(x_i) - \prod_{i=1}^d \tilde{\mathbb{1}}^\delta_{(a_i, b_i)}(x_i) \right| \, dx$$

$$+ \int_{[-M,M]^d} \left| \prod_{i=1}^d \tilde{\mathbb{1}}^\delta_{(a_i, b_i)}(x_i) - \tilde{\times}_d \left( \tilde{\mathbb{1}}^\delta_{(a_1, b_1)}(x_1), \ldots, \tilde{\mathbb{1}}^\delta_{(a_d, b_d)}(x_d) \right) \right| \, dx$$

$$< \int_{\times_{i=1}^d [a_i - \delta, b_i + \delta] \setminus \times_{i=1}^d [a_i, b_i]} \left| \prod_{i=1}^d \tilde{\mathbb{1}}^\delta_{(a_i, b_i)}(x_i) \right| + \int_{[-M,M]^d} \frac{\epsilon}{2^{d+1} M^d} \, dx$$

$$< \text{Vol} \left( \bigtimes_{i=1}^d (a_i - \delta, b_i + \delta) \setminus \bigtimes_{i=1}^d (a_i, b_i) \right) + \frac{\epsilon}{2}$$

where last inequality follows since $\left| \prod_{i=1}^d \tilde{\mathbb{1}}^\delta_{(a_i, b_i)}(x_i) \right| < 1$. We now focus on bounding the measure of $R = \left( \times_{i=1}^d (a_i - \delta, b_i + \delta) \setminus \times_{i=1}^d (a_i, b_i) \right)$. First we express $R = \bigcup_{k=1}^d S_k^+ \cup \bigcup_{k=1}^d S_k^-$ where

$$S_k^- = \left( \bigtimes_{i=1}^{k-1} (a_i - \delta, b_i + \delta) \right) \times (a_k - \delta, a_k) \times \left( \bigtimes_{i=k+1}^d (a_i - \delta, b_i + \delta) \right),$$

$$S_k^+ = \left( \bigtimes_{i=1}^{k-1} (a_i - \delta, b_i + \delta) \right) \times (b_k, b_k + \delta) \times \left( \bigtimes_{i=k+1}^d (a_i - \delta, b_i + \delta) \right).$$

Thus we have that $\text{Vol}(R) \leqslant \sum_{k=1}^{d} \text{Vol}(S_k^+) + \text{Vol}(S_k^-) = 2(2\delta + b_i - a_i)^{d-1}\delta < 2(2\delta + 2M)^{d-1}\delta$. If we pick $\delta = \frac{\epsilon}{4(3M)^{d-1}}$, then we have

$$\|\mathbb{1}_Q - \phi\|_{L^1} < \text{Vol}\left( \bigtimes_{i=1}^{d}(a_i - \delta, b_i + \delta) \setminus \bigtimes_{i=1}^{d}(a_i, b_i) \right) + \frac{\epsilon}{2}$$

$$\leqslant 2(2\delta + 2M)^{d-1}\delta + \frac{\epsilon}{2}$$

$$= 2(3M)^{d-1}\frac{\epsilon}{4(3M)^{d-1}} + \frac{\epsilon}{2}$$

$$= \frac{\epsilon}{2} + \frac{\epsilon}{2} = \epsilon.$$

Finally, we determine the size of $\phi$. Each $\tilde{\mathbb{1}}_{(a_i,b_i)}^{\delta}$ can be implemented by a ReLU network with depth $c_1 \log\left(\frac{1}{\delta}\right) = c_1 \log\left(\frac{4 \cdot 3^{d-1}M^{d-1}}{\epsilon}\right) \leqslant c_2 \log\left(\frac{3^{d-1}M^{d-1}}{\epsilon}\right)$, width 4, and weights bounded by $M$. $\tilde{\times}_d$ can be implemented by a ReLU network with depth at most

$$c_3 \log\left(\frac{d^3 2^d}{\eta}\right) + c_4 = c_3 \log\left(\frac{d^3 2^d 2^{d+1} M^d}{\epsilon}\right) + c_4 = c_3 \log\left(\frac{d^3 2^{2d+1} M^d}{\epsilon}\right) + c_4,$$

width at most $4d$, and weights bounded by 2. Thus $\phi$ has width at most $4d$, weights bounded by $\max\{M, 2\}$, and depth at most

$$c_2 \log\left(\frac{3^{d-1}M^{d-1}}{\epsilon}\right) + c_3 \log\left(\frac{d^3 2^{2d+1} M^d}{\epsilon}\right) + c_4$$

$$< \max(c_2, c_3) \log\left(\frac{d^3 3^{d-1} 2^{2d+1} M^{2d-1}}{\epsilon^2}\right) + c_4$$

$$< \max(c_2, c_3) \log\left(\frac{d^3 4^{d-1} 4^{d+1} M^{2d}}{\epsilon^2}\right) + c_4$$

$$< \max(c_2, c_3) \log\left(\frac{(d^2 4^d M^d)^2}{\epsilon^2}\right) + c_4$$

$$= c_5 \log\left(\frac{d^2 4^d M^d}{\epsilon}\right) + c_4$$

$$= c_5 \log\left(\frac{d^2 4^d}{\epsilon}\right) + c_5 d \log(M) + c_4.$$

$$< c_6 \log\left(\frac{d^2 4^d}{\epsilon}\right) + c_4.$$

$\square$

Now we use the construction of Lemma 10 to approximate the function from Lemma 6 as follows. Let $\{Q_{\vec{k}}\}_{\vec{k} \in [n]^d}$ be a decomposition of $[0, 1]^d$ into almost non-overlapping cubes as in Lemma 6. The only overlap of these cubes is a set of measure 0.

**Lemma 11.** *Let $0 < \epsilon < \frac{1}{2n}$. Let $\{\beta_{\vec{k}}\}_{\vec{k} \in [n]^d}$ be constants within $[-M, M]$. Then the function $g$ defined by*

$$g(x) = \sum_{k \in [n]^d} \beta_{\vec{k}} \mathbb{1}_{Q_{\vec{k}}}(x)$$

*can be approximated by a neural network $\tilde{g}$ with depth $c_1 \log\left(\frac{d^2 4^d n^d}{\epsilon}\right) + c_2$, width $4dn^d$, and weights bounded by $\max\{M, 2\}$ (where $c_1$ and $c_2$ are constants only depending on $M$), such that*

$$\int_{[-M,M]^d} |g(x) - \tilde{g}(x)|\, dx < \epsilon.$$

*Proof.* For every $\vec{k} \in [n]^d$, let $\phi_{\vec{k}}$ be the function from Lemma 10 that approximates $\mathbb{1}_{Q_{\vec{k}}}$ with error $\frac{\epsilon}{Mn^d}$. Then each $\phi_{\vec{k}}$ can be implemented by a ReLU network with

$$c_1 \log\left(\frac{d^2 4^d n^d M}{\epsilon}\right) + c_2 = c_1 \log\left(\frac{d^2 4^d n^d}{\epsilon}\right) + c_3$$

layers, width at most $4d$, and weights bounded by 2. This means we can implement the function $\tilde{g}(x) = \sum_{k \in [n]^d} \beta_{\vec{k}} \phi_{\vec{k}}$ by a ReLU network with one more layer, width at most $4dn^d$, and weights bounded by $\max\{M, 2\}$. We compute

$$
\begin{aligned}
\int_{[-M,M]^d} |g(x) - \tilde{g}(x)|\, dx &= \int_{[-M,M]^d} \left| \sum_{k \in [n]^d} \beta_{\vec{k}} \mathbb{1}_{Q_{\vec{k}}}(x) - \sum_{k \in [n]^d} \beta_{\vec{k}} \phi_{\vec{k}}(x) \right| dx \\
&\leqslant \sum_{k \in [n]^d} |\beta_{\vec{k}}| \int_{[-M,M]^d} \left| \mathbb{1}_{Q_{\vec{k}}}(x) - \phi_{\vec{k}}(x) \right| dx \\
&\leqslant \sum_{k \in [n]^d} |\beta_{\vec{k}}| \left(\frac{\epsilon}{Mn^d}\right) \\
&\leqslant \sum_{k \in [n]^d} M \left(\frac{\epsilon}{Mn^d}\right) = \epsilon.
\end{aligned}
$$

$\square$

## A.3  Putting approximations together

We combine Lemma 6 with Lemma 11 to obtain our approximation result. The requirement of $d \geqslant 4$ in the following lemma is for technical reasons, and is not a requirement for Lemma 3 which is used in the main paper.

**Lemma 12.** *Suppose $f \in C^\alpha([0,1]^d)$, $\alpha \in (0,1)$, with $\|f\|_{L^\infty} < M$ and $M \geqslant 2$. Assume further that $d \geqslant 4$. Let $\epsilon > 0$. Then there exists a function $\Phi$ implementable by a ReLU network such that*

$$\|f - \Phi\|_{L^1} < \epsilon.$$

*The ReLU network has depth at most $c_1 d \log\left(\frac{8d\|f\|_{C^\alpha}^{1/\alpha}}{\epsilon^{2/\alpha}}\right) + c_2$, width at most $\frac{(4d)^{d/2}\|f\|_{C^\alpha}^{d/\alpha}}{\epsilon^{d/\alpha}}$, and weights bounded by $M$ (where $c_1$ and $c_2$ are constants only depending on $M$).*

*Proof.* Let $n = \left\lceil \left(\frac{\|f\|_{C^\alpha}}{\epsilon}\right)^{1/\alpha} \sqrt{d} \right\rceil$. Let $f^n$ be the piecewise constant function from Lemma 6. Notice that $f^n$ follows the same form as the function $g$ in Lemma 11. By Lemma 11, there is a ReLU network function $\Phi$ that approximates $f^n$ such that

$$\int_{[-M,M]^d} |f^n(x) - \Phi(x)|\, dx < \frac{\epsilon}{2}.$$

Then we compute

$$\|f - \Phi\|_{L^1} \leqslant \|f - f^n\|_{L^1} + \|f^n - \Phi\|_{L^1} < \frac{\|f\|_{C^\alpha} d^{\alpha/2}}{n^\alpha} + \frac{\epsilon}{2} < \frac{\epsilon}{2} + \frac{\epsilon}{2} = \epsilon.$$

Note that $\Phi$ has width $4dn^d < 4d\left(2\left(\frac{\|f\|_{C^\alpha}}{\epsilon}\right)^{1/\alpha}\sqrt{d}\right)^d = c_0\frac{(4d)^{d/2}\|f\|_{C^\alpha}^{d/\alpha}}{\epsilon^{d/\alpha}}$, and the weights of $\Phi$ are bounded by $M$. The depth of $\Phi$ is bounded by

$$c_1\log\left(\frac{d^2 4^d n^d}{\epsilon}\right) + c_2 < c_1\log\left(\frac{d^2 4^d \|f\|_{C^\alpha}^{\frac{d}{\alpha}} 2^d d^{\frac{d}{2}}}{\epsilon\epsilon^{\frac{d}{\alpha}}}\right) + c_2$$

$$< c_1\log\left(\frac{d^d 8^d \|f\|_{C^\alpha}^{\frac{d}{\alpha}}}{\epsilon^{\frac{2d}{\alpha}}}\right) + c_2$$

$$= c_1 d\log\left(\frac{8d\|f\|_{C^\alpha}^{1/\alpha}}{\epsilon^{2/\alpha}}\right) + c_2$$

where we use the fact that $d \geqslant 4$ to bound $2 + \frac{d}{2} \leqslant d$.

$\square$

In the main paper, we use Lemma 3, which is proved below.

*Proof of Lemma 3.* By allowing the constants to depend on all values except $\epsilon$, the depth of the ReLU network $\Phi$ from Lemma 12 can be expressed as

$$c_1 d\log\left(\frac{8d\|f\|_{C^\alpha}^{1/\alpha}}{\epsilon^{2/\alpha}}\right) + c_2 = c_3\left(\log\left(\frac{1}{\epsilon^{2/\alpha}}\right) + \log\left(8d\|f\|_{C^\alpha}^{1/\alpha}\right)\right) + c_2 = \frac{2}{\alpha}c_3\log\left(\frac{1}{\epsilon}\right) + c_4 < c_5\log\left(\frac{1}{\epsilon}\right),$$

and the width can be expressed as

$$\frac{(4d)^{d/2}\|f\|_{C^\alpha}^{d/\alpha}}{\epsilon^{d/\alpha}} = \frac{c_6}{\epsilon^{d/\alpha}}.$$

$\square$

## A.4 Approximating $g^*$ in (10) with a neural network

The goal is to show that the $L^1$ distance between $g_\theta$ (as defined in 11) and $g^*$ is small. We have

$$\|g^* - g_\theta\|_{L^1}$$

$$= \sum_{i=1}^{D}\|(g^*)^{(i)} - (g_\theta)^{(i)}\|_{L^1}$$

$$= \sum_{i=1}^{D}\int_{(0,1)^{d+1}}\left|(g^*)^{(i)}(x) - (g_\theta)^{(i)}(x)\right|dx$$

$$\leqslant \sum_{i=1}^{D}\sum_{j=1}^{J}\int_{(0,1)^{d+1}}\left|\tilde{\times}^{\delta_2}\left(\tilde{\mathbb{1}}^{\delta_1}_{(\pi_{j-1},\pi_j)}(x_1),(g^{\delta_3}_{j,\theta})^{(i)}(x_{2:d+1})\right) - \mathbb{1}_{(\pi_{j-1},\pi_j)}(x_1)(g^*_j)^{(i)}(x_{2:d+1})\right|dx$$

$$\leqslant \sum_{i=1}^{D}\sum_{j=1}^{J}\int_{(0,1)^{d+1}}\left|\tilde{\times}^{\delta_2}\left(\tilde{\mathbb{1}}^{\delta_1}_{(\pi_{j-1},\pi_j)}(x_1),(g^{\delta_3}_{j,\theta})^{(i)}(x_{2:d+1})\right) - \tilde{\mathbb{1}}^{\delta_1}_{(\pi_{j-1},\pi_j)}(x_1)(g^{\delta_3}_{j,\theta})^{(i)}(x_{2:d+1})\right|dx$$

$$+ \sum_{i=1}^{D}\sum_{j=1}^{J}\int_{(0,1)^{d+1}}\left|\tilde{\mathbb{1}}^{\delta_1}_{(\pi_{j-1},\pi_j)}(x_1)(g^{\delta_3}_{j,\theta})^{(i)}(x_{2:d+1}) - \mathbb{1}_{(\pi_{j-1},\pi_j)}(x_1)(g^{\delta_3}_{j,\theta})^{(i)}(x_{2:d+1})\right|dx$$

$$+ \sum_{i=1}^{D}\sum_{j=1}^{J}\int_{(0,1)^{d+1}}\left|\mathbb{1}_{(\pi_{j-1},\pi_j)}(x_1)(g^{\delta_3}_{j,\theta})^{(i)}(x_{2:d+1}) - \mathbb{1}_{(\pi_{j-1},\pi_j)}(x_1)(g^*_j)^{(i)}(x_{2:d+1})\right|dx$$

$$= \sum_{i=1}^{D}\sum_{j=1}^{J}\left((\text{I}) + (\text{II}) + (\text{III})\right)$$

Each of the three terms are easily handled as follows.

(I) By construction of $\tilde{\times}^{\delta_2}$ in Lemma 7, we have that

$$
\begin{aligned}
\text{(I)} &= \int_{(0,1)^{d+1}} \left| \tilde{\times}^{\delta_2}\left( \tilde{\mathbb{1}}^{\delta_1}_{(\pi_{j-1},\pi_j)}(x_1), (g^{\delta_3}_{j,\theta})^{(i)}(x_{2:d+1}) \right) - \tilde{\mathbb{1}}^{\delta_1}_{(\pi_{j-1},\pi_j)}(x_1)(g^{\delta_3}_{j,\theta})^{(i)}(x_{2:d+1}) \right| dx \\
&\leqslant \int_{(0,1)^{d+1}} \delta_2 \, dx = \delta_2.
\end{aligned}
$$

(II) By construction of $\tilde{\mathbb{1}}^{\delta_1}_{(\pi_{j-1},\pi_j)}$ in Lemma 9, we have that

$$
\begin{aligned}
\text{(II)} &= \int_{(0,1)^{d+1}} \left| \tilde{\mathbb{1}}^{\delta_1}_{(\pi_{j-1},\pi_j)}(x_1)(g^{\delta_3}_{j,\theta})^{(i)}(x_{2:d+1}) - \mathbb{1}_{(\pi_{j-1},\pi_j)}(x_1)(g^{\delta_3}_{j,\theta})^{(i)}(x_{2:d+1}) \right| dx \\
&\leqslant \left\| (g^{\delta_3}_{j,\theta})^{(i)} \right\|_\infty \int_0^1 \left| \tilde{\mathbb{1}}^{\delta_1}_{(\pi_{j-1},\pi_j)}(x_1) - \mathbb{1}_{(\pi_{j-1},\pi_j)}(x_1) \right| dx \\
&\leqslant M \left\| \tilde{\mathbb{1}}^{\delta_1}_{(a,b)} - \mathbb{1}_{(a,b)} \right\|_{L^1} \\
&= M\delta_1.
\end{aligned}
$$

(III) By construction of $(g^{\delta_3}_{j,\theta})^{(i)}$ from Lemma 3, we have that

$$
\begin{aligned}
\text{(III)} &= \int_{(0,1)^{d+1}} \left| \mathbb{1}_{(\pi_{j-1},\pi_j)}(x_1)(g^{\delta_3}_{j,\theta})^{(i)}(x_{2:d+1}) - \mathbb{1}_{(\pi_{j-1},\pi_j)}(x_1)(g^*_j)^{(i)}(x_{2:d+1}) \right| dx \\
&= \| \mathbb{1}_{(\pi_{j-1},\pi_j)} \|_\infty \int_{(0,1)^d} \left| (g^{\delta_3}_{j,\theta})^{(i)}(x) - (g^*_j)^{(i)}(x) \right| dx \\
&= \left\| (g^{\delta_3}_{j,\theta})^{(i)} - (g^*_j)^{(i)} \right\|_{L^1} \\
&\leqslant \delta_3.
\end{aligned}
$$

As a result, we have that

$$
\| g^* - g_\theta \|_{L^1} \leqslant \sum_{i=1}^D \sum_{j=1}^J \text{(I)} + \text{(II)} + \text{(III)} \leqslant \sum_{i=1}^D \sum_{j=1}^J \delta_2 + M\delta_1 + \delta_3 = DJ(M\delta_1 + \delta_2 + \delta_3).
$$

By selecting $\delta_1 < \frac{\epsilon}{3DJM}$, $\delta_2 < \frac{\epsilon}{3DJ}$, and $\delta_3 < \frac{\epsilon}{3DJ}$, we obtain that $\| g^* - g_\theta \|_1 < \epsilon$.

To complete the proof, we note that $g_\theta$ can be exactly represented by a neural network in $\mathcal{G}_{\text{NN}}(L, p, \kappa)$ with parameters

$$
L = O\left( \log\left( \frac{1}{\epsilon} \right) \right), \quad p = O\left( D\epsilon^{-\frac{d}{\alpha}} \right), \quad \kappa = M.
$$

## B    Statistical Lemmas

Here we present proofs of lemmas used in our statistical theory in Section 4.2.

### B.1    Distribution approximation in $W_1$ via function approximation in $L^1$

*Proof of Lemma 2.* The vector-valued functions $f$ and $g$ output $m$-dimensional vectors. Note that $\| f - g \|_{L^1(\mu)} = \sum_{i=1}^m \| f_i - g_i \|$ where $f_i$ and $g_i$ denote the $i$th component function of $f$ and $g$,

respectively. Then we can compute

$$
\begin{aligned}
W_1(f_\sharp\mu, g_\sharp\mu) &= \sup_{\phi\in\mathrm{Lip}_1(\mathbb{R}^m)} \left| \int \phi(y)\, d(f_\sharp\mu) - \int \phi(y)\, d(g_\sharp\mu) \right| \\
&= \sup_{\phi\in\mathrm{Lip}_1(\mathbb{R}^m)} \left| \int \phi(f(x)) - \phi(g(x))\, d\mu \right| \\
&\leqslant \sup_{\phi\in\mathrm{Lip}_1(\mathbb{R}^m)} \int |\phi(f(x)) - \phi(g(x))|\, d\mu \\
&\leqslant \int_Z \|f(x) - g(x)\|_2\, d\mu \\
&\leqslant \int_Z C\|f(x) - g(x)\|_1\, d\mu \\
&= C\|f - g\|_{L^1(\mu)},
\end{aligned}
$$

since $\phi$ is Lipschitz with constant 1 and all norms are equivalent in finite dimensions. In particular, $C = 1$ here.

$\square$

## B.2 Convergence of empirical measure

*Proof of Lemma 5.* Let $\delta > 0$. Consider the manifold $\mathcal{M}$ with the geodesic distance as a metric space. When [Weed and Bach, 2019, Theorem 1] is applied to $\mathcal{M}$ with the geodesic distance, we have that

$$
\mathbb{E}\left[W_1^{\mathcal{M}}(Q, Q_n)\right] \leqslant C_\delta n^{-\frac{1}{d+\delta}}
$$

for some constant $C_\delta$ independent of $n$. Here, $W_1^{\mathcal{M}}$ is the 1-Wasserstein distance on $\mathcal{M}$ with the geodesic distance. It suffices to show that

$$
W_1^{\mathbb{R}^D}(Q, Q_n) = W_1(Q, Q_n) \leqslant W_1^{\mathcal{M}}(Q, Q_n).
$$

Let $\mathrm{Lip}_1(\mathbb{R}^D)$ and $\mathrm{Lip}_1(\mathcal{M})$ denote the set of 1-Lipschitz functions defined on $\mathcal{M}$ with respect to the Euclidean distance on $\mathbb{R}^D$ and geodesic distance on $\mathcal{M}$ respectively. But note that $\mathrm{Lip}_1(\mathbb{R}^D) \subseteq \mathrm{Lip}_1(\mathcal{M})$ because for any $f \in \mathrm{Lip}_1(\mathbb{R}^D)$ we have

$$
\frac{|f(x) - f(y)|}{\|x - y\|_{\mathcal{M}}} \leqslant \frac{|f(x) - f(y)|}{\|x - y\|_{\mathbb{R}^D}} \leqslant 1
$$

as $\|x - y\|_{\mathbb{R}^D} \leqslant \|x - y\|_{\mathcal{M}}$ under an isometric embedding and hence $f \in \mathrm{Lip}_1(\mathcal{M})$. Thus

$$
\mathbb{E}\left[W_1(Q, Q_n)\right] \leqslant \mathbb{E}\left[W_1^{\mathcal{M}}(Q, Q_n)\right] \leqslant C_\delta n^{-\frac{1}{d+\delta}}.
$$

$\square$

## B.3 Controlling the noisy samples

In the noisy setting, we are given $n$ noisy i.i.d. samples $\hat{X}_1, ..., \hat{X}_n$ of the form $\hat{X}_i = X_i + \xi_i$, for $X_i \sim Q$ and $\xi_i$ distributed according to some noise distribution. The optimization in (4) is performed with the noisy empirical distribution $\hat{Q}_n = \frac{1}{n}\sum_{i=1}^n \delta_{\hat{X}_i}$.

**Lemma 13.** *Under the same assumptions of Theorem 2 and in the noisy setting, we have*

$$
\mathbb{E}W_1((\hat{g}_n)_\sharp\rho, Q) \leqslant \inf_{g_\theta\in\mathcal{G}_{\mathrm{NN}}} W_1((g_\theta)_\sharp\rho, Q) + 2\mathbb{E}W_1(Q_n, Q) + 2\mathbb{E}W_1(\hat{Q}_n, Q_n) \tag{15}
$$

*where $\hat{Q}_n$ is the noisy empirical distribution and $Q_n$ is the clean empirical distribution.*

*Proof.* We compute recalling the definition of $\hat{g}_n$ as the empirical risk minimizer.

$$
\begin{aligned}
\mathbb{E}W_1((\hat{g}_n)_\sharp\rho, Q) &\leqslant \mathbb{E}W_1((\hat{g}_n)_\sharp\rho, \hat{Q}_n) + \mathbb{E}W_1(\hat{Q}_n, Q) \\
&\leqslant \mathbb{E}\inf_{g_\theta\in\mathcal{G}_{\mathrm{NN}}} W_1((g_\theta)_\sharp\rho, \hat{Q}_n) + \mathbb{E}W_1(Q_n, Q) + \mathbb{E}W_1(\hat{Q}_n, Q_n) \\
&\leqslant \mathbb{E}\inf_{g_\theta\in\mathcal{G}_{\mathrm{NN}}} W_1((g_\theta)_\sharp\rho, Q) + 2\mathbb{E}W_1(Q_n, Q) + 2\mathbb{E}W_1(\hat{Q}_n, Q)
\end{aligned}
$$

where where we recall $W_1((\hat{g}_n)_\sharp\rho, \hat{Q}_n) = \inf_{g_\theta\in\mathcal{G}_{\mathrm{NN}}} W_1((g_\theta)_\sharp\rho, \hat{Q}_n)$ from (4).

$\square$

**Lemma 14.** *Write* $W_1(\hat{Q}_n, Q_n) = W_1^{\mathbb{R}^D}(\hat{Q}_n, Q_n)$. *In the noisy setting, we express* $\hat{X}_i = X_i + \xi_i$ *where* $X_i$ *is drawn from* $Q$ *and then noised with* $\xi_i$ *drawn from some noise distribution. Then*

$$\mathbb{E}[W_1(Q_n, \hat{Q}_n)] \leqslant \sqrt{V_\xi}$$

*where* $V_\xi = \mathbb{E}\|\xi\|_2^2$ *which is the variance of the noise.*

*Proof.* Let $\hat{X}_i, X_i$ be samples defining $\hat{Q}_n, Q_n$ respectively. We have $\hat{X}_i = X_i + \xi_i$ where $\xi$ is the noise term. Compute

$$\mathbb{E}W_1(\hat{Q}_n, Q_n) = \mathbb{E}\sup_{f\in\mathrm{Lip}_1(\mathbb{R}^D)} \hat{Q}_n(f) - Q_n(f) = \mathbb{E}\sup_{f\in\mathrm{Lip}_1(\mathbb{R}^D)} \frac{1}{n}\sum_{i=1}^n f(\hat{X}_i) - f(X_i)$$

$$\leqslant \mathbb{E}\sup_{f\in\mathrm{Lip}_1(\mathbb{R}^D)} \frac{1}{n}\sum_{i=1}^n |f(\hat{X}_i) - f(X_i)| = \mathbb{E}\sup_{f\in\mathrm{Lip}_1(\mathbb{R}^D)} \frac{1}{n}\sum_{i=1}^n |f(X_i + \xi_i) - f(X_i)|$$

$$\leqslant \mathbb{E}\sup_{f\in\mathrm{Lip}_1(\mathbb{R}^D)} \frac{1}{n}\sum_{i=1}^n \|\xi_i\|_2 = \mathbb{E}\|\xi\|_2 \leqslant \sqrt{V_\xi}$$

the last line follows from Jensen's inequality. $\square$

We conclude in the noisy setting that

$$\mathbb{E}W_1((\hat{g}_n)_\sharp\rho, Q) \leqslant \epsilon_{\mathrm{appx}} + 2C_\delta n^{-\frac{1}{d+\delta}} + 2\sqrt{V} \leqslant Cn^{-\frac{1}{d+\delta}} + 2\sqrt{V_\xi}$$

after balancing the approximation error $\epsilon_{\mathrm{appx}}$ appropriately.