# OpenReview forum: "On Deep Generative Models for Approximation and Estimation of Distributions on Manifolds"
_NeurIPS.cc/2022/Conference — NeurIPS 2022 Accept_

### Official Review · Reviewer_ufvV · 2022-07-11

**Rating:** 7
**Confidence:** 3
**Soundness:** 3 good
**Presentation:** 3 good
**Contribution:** 3 good

**Summary:**

This paper proves approximation and estimation theories of deep generative networks for estimating distributions on a low-dimensional manifold under the Wasserstein-1 loss.

The assumptions on the manifold are standard and weaker than related work. In particular, it does not assume that the manifold is globally homeomorphic to Euclidean space.

The assumptions on the target data distribution are different from (weaker than?) the related work. It requires that the density is both upper and lower bounded without smoothness assumptions.

The resulting theories depend on the intrinsic dimension of the manifold instead of the dimension of the data.

**Questions:**

A minor thing: C in Assumption 2 and that in Theorem 2 have the same meaning?

**Limitations:**

I did not see a direct negative societal impact of the work since it is purely theoretical.

**Strengths And Weaknesses:**

Strengths

1. The paper is clearly written and the related work is properly discussed.

2. The problem is important and the technique is novel.

Weakness

1. The authors may discuss the bounded assumptions on the distribution. Does previous work make a similar one? How do C and c in Assumption 2 affect the results?

---

> ### Author Response · Authors · 2022-08-02
> **Author Rebuttal**
>
> Thank you for your review. Here are our responses.
>
> ## Weaknesses
>
> 1. **The authors may discuss the bounded assumptions on the distribution. Does previous work make a similar one? How do C and c in Assumption 2 affect the results?**
>
>     A crucial ingredient of our proof is from the optimal transport theory which requires boundedness of the density. This assumption is necessary to guarantee regularity of the optimal transport map. Also, see our response to Weakness #4 by Reviewer P4YA. The constants C and c in Assumption 2 are absorbed by the big-O notation in our results.
>
>
> ## Questions
> 1. **A minor thing: C in Assumption 2 and that in Theorem 2 have the same meaning?**
>
>    The C in Assumption 2 and the C in Theorem 2 are different constants. This will specified in a further revision.

---

### Official Review · Reviewer_Q3cg · 2022-07-11

**Rating:** 6
**Confidence:** 1
**Soundness:** 3 good
**Presentation:** 3 good
**Contribution:** 3 good

**Summary:**

The authors perform a theoretical analysis of the ability of deep neural networks to map an easy-to-sample distribution to a distribution supported on a low dimensional manifold embedded in a high-dimensional space.  They have two main theorems.  The first theorem identifies that there exists a Holder-continuous transport map that transforms the easy-to-sample distribution to a distribution supported on the low dimensional manifold.  Then it identifies that, for any epsilon>0, there exists a deep neural network that transforms the easy-to-sample distribution to a distribution that is epsilon-close to the desired manifold distribution in the Wasserstein-1 distance. The second theorem considers the case where there are only n samples of the target distribution.  It shows that, for any delta>0, one can design a deep neural network for which the expected Wasserstein-1 distance is upper bounded by a constant times n^{1/(d+delta)}, where d is the manifold dimension. For these results, the manifold is assumed to be a compact Riemannian manifold with a positive reach, the target distribution's density is assumed to be bounded from above and below, and the network is assumed to alternate between affine linear layers and ReLU layers.

**Questions:**

As a matter of terminology, it's not clear how "the network size is exponential in the intrinsic dimension d".  To me that would mean that some characterization of size (e.g., width) grows proportional to exp(d) or perhaps exp(d/alpha).  But the results show that the width grows as D*exp(-d/alpha).

On line 49, the authors assert that existing results claim that "the sample size is required to be epsilon^{-D}" after which they plug in particular values for epsilon and D.  But I don't think any existing results claim exactly this many samples are required.  Rather they claim sample complexity that grows proportional to epsilon^{-D}, which prevents assigning any particular number of samples.

I have trouble with the interpretation of "reach" after Definition 3 and the sphere example.  To satisfy Definition 3, why can't we take two arbitrarily close points on the sphere and then a third point on the plane bisecting them, but off the sphere? Is the reach of the sphere really its radius according to Definition 3?

**Limitations:**

I don't see any potential for negative societal impact.

**Strengths And Weaknesses:**

The theory confirms what everybody knows from 1000s of GAN papers: generative neural nets can successfully approximate distributions on manifolds. I imagine that the results are of interest to the theoretical community, but I don't have sufficient expertise to say whether the proofs are completely sound or whether the results are as novel as suggested by the authors.

As for weaknesses, it's not obvious that the results will have an impact on the practice of machine learning. For example, do they suggest any changes to how we currently design or train GANs?

And it's not clear how sharp the bounds are.  Is it likely that they will be improved in future work?  Do they agree with experiments?

---

> ### Author Response · Authors · 2022-08-02
> **Author Rebuttal**
>
> Thank you for the review. Here are our responses.
>
> ## Weaknesses
>
> 1. **It's not obvious that the results will have an impact on the practice of machine learning. For example, do they suggest any changes to how we currently design or train GANs?**
>
>      This result provides a theoretical justification, through mathematical proofs, for the well-accepted and widely-observed capability of GANs to leverage intrinsically low-dimensional data structures.
>
>
> 2. **It is not clear how sharp the bounds are. Is it likely that they will be improved in future work? Do they agree with experiments?**
>
>     We believe that our bound is tight in the dependence on the intrinsic dimension $d$. Our current theory does not require any smoothness assumption on the density of the data distribution. If we put additional assumptions on the regularity of the density we could improve bounds in future works.
>
> ## Questions
>
> 1. **As a matter of terminology, it's not clear how "the network size is exponential in the intrinsic dimension d". To me that would mean that some characterization of size (e.g., width) grows proportional to exp(d) or perhaps exp(d/alpha). But the results show that the width grows as D*exp(-d/alpha).**
>
>     We say exponential in the intrinsic dimension when we write $(\epsilon^{-d/\alpha})$ in Theorem 1 where $\epsilon$ is the desired accuracy. In Theorem 2, we choose $\epsilon$ to be a small number as Line 322. We want to clarify that $\epsilon$ is not $\exp$ in our paper.
>
>
> 2. **On line 49, the authors assert that existing results claim that "the sample size is required to be $\epsilon^{-D}$" after which they plug in particular values for epsilon and D. But I don't think any existing results claim exactly this many samples are required. Rather they claim sample complexity that grows proportional to $\epsilon^{-D}$, which prevents assigning any particular number of samples.**
>
>     We agree we may not require exactly this many samples in practice, but rather theory suggests we will need this order of samples when the accuracy $\epsilon$ is sufficiently small. For example, see [Theorem 3.1](https://arxiv.org/pdf/1712.08244.pdf) and [Theorem 1 and Theorem 2](https://arxiv.org/abs/2002.03938v2).
>
>
> 3. **I have trouble with the interpretation of "reach" after Definition 3 and the sphere example. To satisfy Definition 3, why can't we take two arbitrarily close points on the sphere and then a third point on the plane bisecting them, but off the sphere? Is the reach of the sphere really its radius according to Definition 3?**
>
>     Thanks for pointing this out! In fact the definition has a typo and we should require the distance stated must be the minimum distance from the sphere. Formally $\tau = \inf \lbrace r > 0 : \exists x\neq y \in \mathcal{M}, v \in \mathbb{R}^D \textup{ such that } r =\|x- v\| = \|y- v\| = \inf_{z \in \mathcal{M}} \| z - v \| \rbrace$. In this case the reach should be the radius of the sphere.

---

### Official Review · Reviewer_CSiC · 2022-07-11

**Rating:** 7
**Confidence:** 3
**Soundness:** 3 good
**Presentation:** 3 good
**Contribution:** 3 good

**Summary:**

This paper presents a theory of generative neural networks when it is assumed that the real-world data studied lies on a low-dimensional manifold. This is a common hypothesis in machine learning, to try and reconcile the very high dimensionality of most datasets with the effectiveness of classical machine learning methods, such as GANs.
Assuming that the data lies on a $d$-dimensional compact manifold, the authors show that for a specific GAN architecture, trained with the Wasserstein loss, the approximation error between the actual distribution $Q$ and the GAN measure $\mu_n$ trained by ERM on $n$ samples reads
$$ W_1(Q, \mu_n) \leq C n ^{1/d}, $$
independently of the embedding distribution $D$. The curse of dimensionality thus only applies for the intrisic dimension of the data, and not the ambient one.

The proof relies on partitioning the manifold $\cal M$ into finitely many disjoint (Voronoi) cells, that can be mapped to subsets of euclidean space via diffeomorphisms. Then, those subsets are mapped onto open balls, so that the distribution $Q$ is the pushforward of the uniform measure on a finite union of disjoint open balls in $\mathbb R^d$. This can be simplified further to the uniform measure on $[0, 1]^{d+1}$, using the last coordinate to choose which ball to sample from. The results then follow from classical theorems on ERM for the Wasserstein distance.

**Questions:**

- what happens when the support of $Q$ is not compact, but still lies on a compact manifold?
- for Lemma 2, why isn't the map
$$ x \mapsto \frac{Lx}{R(x/\lVert x \rVert)} $$
sufficient, since  $R$ is lipschitz and bounded from below ?

Geometry questions/remarks:
- Definition 1 uses the concept of *geodesic* but without specifying its second endpoint; it might be helpful to explain why this is the case (i.e. how to intrisically define geodesics)
- Is equation (5) only valid for the exponential map ? If that is not the case, why choose the exponential map instead of any atlas ?
- the notion of *reach* of a manifold is ill-defined: it lacks the requirement that $r$ is the minimal distance from $v$ to $\cal M$.



**Limitations:**

I don't really agree that any paper working on generative models can claim not having "any negative social impact"; however, the potential risks of generative models are already quite well described in the literature.

**Strengths And Weaknesses:**

The mathematical study of how neural networks break the curse of dimensionality is a very interesting and challenging topic, with few rigorous results. This paper manages to show a very general bound for ERM, that depends only on the structure of the underlying manifold instead of the ambient space. The assumptions on the manifold $\cal M$ and the distribution $Q$ are quite lax, much more than the usual Gaussian features assumption (which would roughly correspond to the uniform distribution on the $d$-sphere). The main drawback, result-wise, is that the architecture of the resulting GAN is a priori highly dependent on the manifold structure; it would have been insightful to see if a general GAN with this structure can fit real or synthetic data with the proclaimed accuracy.

On the presentation side, the article is quite heavy with riemannian geometry concepts, which might not be familiar to most Neurips readers. It is therefore important that those concepts be clearly introduced, and for the most part this is nicely done. Some unclear points remain, which are categorized in the following section. The proof structure is clearly marked, with helpful figures providing additional clarity to the arguments. Overall, this is an interesting and insightful paper, somewhat hindered by the wealth of geometry notions to introduce in 10 pages, but still enjoyable to read.

---

> ### Author Response · Authors · 2022-08-02
> **Author Rebuttal**
>
> Thank you for your review. Here are our responses.
>
> ## Questions
> 1. **What happens when the support of $Q$ is not compact, but still lies on a compact manifold?**
>
>     If the support of $Q$ is not a compact manifold, then it is not supported by our theory. Without further assumptions, we cannot use many crucial steps of our proof. For example, the exponential map may not be continuous (or even well-defined) which is essential to our argument. The exponential map is the natural bridge connecting geodesics on the manifold to straight lines in the $d$-dimensional Euclidean space (i.e. via tangent vectors on the manifold), so it essential that it is well-defined for our argument.
>
> 2. **For Lemma 2, why isn't the map $ x \rightarrow \frac{Lx}{R(x / \| x \| )} $ sufficient, since $R$ is lipschitz and bounded from below?**
>
>     Unfortunately, this map is not Lipschitz as the reciprocal-norm function $1/\|x\|$ is only Lipschitz on $(c, \infty)$ for any $c > 0$. In other words, we must keep $x$ away from $0$ for that function, which naturally leads to the construction we use in Lemma 2.
>
> 3. **Definition 1 uses the concept of geodesic but without specifying its second endpoint; it might be helpful to explain why this is the case (i.e. how to intrinsically define geodesics)**
>
>     The notion of a geodesic can be defined equivalently either by specifying an initial point and velocity (as we do) or by specifying an initial and final point with a length minimizing requirement. For further details we refer to chapter 3 of [this classic reference on Riemannian geometry](https://link.springer.com/book/9780817634902).
>
> 4. **Is equation (5) only valid for the exponential map ? If that is not the case, why choose the exponential map instead of any atlas ?**
>
>     The change of variables formula is generally valid for local coordinate maps. However we use the exponential map because we use it to guarantee that the resulting Voronoi cells in the tangent space are star-shaped, which is important in our argument. Furthermore, the exponential map has very natural interpretations in the tangent space as it maps tangent vectors to geodesics.
>
> 5. **The notion of *reach* of a manifold is ill-defined: it lacks the requirement that $r$ is the minimal distance from $v$ to $\mathcal{M}$.**
>
>     Thanks for pointing out the typo! We have fixed it in the updated version.

---

### Official Review · Reviewer_P4YA · 2022-07-11

**Rating:** 7
**Confidence:** 3
**Soundness:** 4 excellent
**Presentation:** 3 good
**Contribution:** 4 excellent

**Summary:**

This paper studies the ability of deep generative models to approximate manifold-supported distributions. In particular, this work shows that generative models can indeed approximate these distributions to high accuracy (in Wasserstein-1 sense), and then shows that the statistical rate of approximation depends on the low intrinsic dimension as opposed to the high ambient dimension. The authors consider a fairly general setting wherein the target manifold is composed of an unspecified number of charts, and the target distribution only has its density (with respect to the restriction of the Hausdorff measure to the manifold) bounded above and below by some constants. The work is purely theoretical.

**Questions:**

1. Does this theory provide any new experimental insight that was not known before? How does this result help practitioners moving forward?
2. Why is the base distribution uniform?
3. How do the assumptions on the manifold and corresponding density relate to real-world problems?

**Limitations:**

This paper is short on acknowledging its own limitations, namely that there is not experimental recommendation or direct benefit to practitioners emerging from this theory. I would appreciate it if the authors mentioned something about this.

**Strengths And Weaknesses:**

## Overall Assessment

Altogether, I find this paper to be very impressive, although I have to admit that I did not have time to go through the proofs thoroughly and may have missed some mistakes. Assuming that everything is correct, though, this paper is very timely and significant as more and more works make the manifold hypothesis an important modelling assumption when performing deep generative modelling. However, I would have appreciated some more discussion on the broader impact associated with these results, and perhaps a bit more intuition on how the results are shown. I'll elaborate below.

## Strengths

Overall, the contributions made by this paper appear to be quite impressive. I found the two research questions to be important to ask, and the answers to these questions help explain phenomena observed by practitioners of deep generative modelling for years. The proofs themselves appear highly nontrivial.

The writing of the paper is generally good. I found the introduction provides a solid motivation for their results and gives a good overview of the paper. The authors also do a good job of making the paper self-contained.

Originality and quality are both very strong in this paper, while both clarity and significance are reasonably strong.

## Weaknesses

The most obvious weakness of this paper is the lack of experimental insight or direct verification of the theoretical results, which takes away from some of the significance of the results. I appreciate that past work has shown that deep generative models can do a decent job of fitting image-like data that lives in high-dimensional space, and this may be considered as experimental verification of the theoretical claims. However, I think it's possible to do more here:
1. It would be interesting to see an experiment wherein the generative process is controlled, such that, for example you can see how much the Wasserstein-1 distance decreases as a function of the number of observed samples.
2. At a higher level, it would be nice to have some new practical insight emerging from the theory that could then itself be tested. I'm not sure what that would be, and unfortunately the paper provides no guidance in this direction.

The next thing I think is a bit weak is the intuition about certain choices made in the paper. For example:
1. Why is the base, easy-to-sample distribution here a uniform on $d+1$ dimensions? Both the dimensionality ($d+1$) and uniform-ness are non-standard choices. I can see later that one of the dimensions is used as an indicator and the others are used for "indexing" the manifold in some sense, but I think an opportunity was missed to provide a bit more discussion on this. The non-Gaussian base distribution should also be expanded on a bit more - do you use a uniform instead of a Gaussian because that allows approximating a distribution with density bounded below away from $0$? Or is it used because $\mathcal M$ is compact?
2. Can we get some more discussion about the assumption that requires the groundtruth density to be bounded away from $0$, or perhaps some examples of distributions which _do not_ satisfy this requirement?

Generally, I think the theoretical results can be bolstered by further probing the assumptions as problem setup, rather than just dismissing these choices as "minimal". It would be nice to discuss how these choices relate to the problems considered by practitioners. The lack of discussion here does take away from the clarity a bit.

---

> ### Author Response · Authors · 2022-08-02
> **Author Rebuttal**
>
> Thank you for your review. Here are our responses.
>
> ## Weaknesses
>
> 1. **It would be interesting to see an experiment wherein the generative process is controlled, such that, for example, you can see how much the Wasserstein-1 distance decreases as a function of the number of observed samples.**
>
>     Thank you for the suggestion. This paper focuses on approximation and generalization theories. We may not be able to provide such simulations now, but we can try it in the revision.
>
> 2. **At a higher level, it would be nice to have some new practical insight emerging from the theory that could then itself be tested. I'm not sure what that would be, and unfortunately the paper provides no guidance in this direction.**
>
>     The goal of this paper is to perform a theoretical analysis to validate the common belief that, when high-dimensional data exhibit low-dimensional structures, one can generate the data distribution from a low-dimensional easy-to-sample distribution. This phenomenon has been extensively observed in literature. Numerical experiments are not necessary here to prove the validity of the results. On the other hand, we do agree that the experiment suggested in Weakness #1 would be interesting, and we are considering it in the revision of this paper.
>
> 3. **Why is the base, easy-to-sample distribution here a uniform on $d + 1$ dimensions? Both the dimensionality ($d + 1$) and uniform-ness are non-standard choices [question truncated due to character limit] ... Or is it used because $\mathcal{M}$ is compact?**
>
>     The manifold $\mathcal{M}$ is covered by multiple charts. To represent the data distribution on the manifold, we use one dimension of the input distribution $\rho$ to select the chart, and the other $d$ dimensions are used to generate local distributions on each chart.
>
>     The “easy-to-sample” distribution $\rho$ can be general in our framework. The only restrictions on $\rho$ come from optimal transport theory, which requires the source distribution to be supported on an open, connected, bounded set with density lower and upper bounded.
> In this paper, we simply choose uniform distribution for ease of presentation. In practice, we could take a truncated and renormalized Gaussian distribution as our easy-to-sample distribution.
>
> 4. **Can we get some more discussion about the assumption that requires the groundtruth density to be bounded away from $0$, or perhaps some examples of distributions which do not satisfy this requirement?**
>
>     We require the density to be bounded away from zero in order to satisfy the assumptions of the optimal transport theory that we crucially use (Proposition 1). More details can be found in Section 1.7.6 of this [reference on optimal transport](http://math.univ-lyon1.fr/~santambrogio/OTAM-cvgmt.pdf). This assumption is necessary to guarantee the regularity of the optimal transport map. An example of a density with bounded support that is not lower bounded away from 0 is the beta distribution for many parameter choices (e.g. $\alpha = \beta = 2$). On the other hand, if the density of the data distribution approaches zero somewhere, it is plausible to neglect a small region around it without sacrificing the distribution approximation and estimation guarantees, but additional technical arguments are needed to make this analysis rigorous.
>
> ## Questions
>
> 1. **Does this theory provide any new experimental insight that was not known before? How does this result help practitioners moving forward?**
>
>     This result provides a theoretical justification, through mathematical proofs, for the well-accepted and widely-observed capability of GANs to leverage intrinsically low-dimensional data structures.
>
> 2. **Why is the base distribution uniform?**
>
>     See our response to Weakness #3 above,
>
> 3. **How do the assumptions on the manifold and corresponding density relate to real-world problems?**
>
>     It has been observed in literature that many well-known datasets exhibit intrinsically low-dimensional structures. For example, it has been verified in [this study on image data](https://openreview.net/forum?id=XJk19XzGq2J) that the intrinsic dimension of the well-known natural image datasets are much smaller than the ambient dimension. This is the key observation that our manifold assumption is based on. Further our assumptions on the manifold are commonly used in the manifold learning literature. For more details, see our response to Weakness #1 by Reviewer Rb3G. We make very weak assumptions on the density, only requiring it to be both upper and lower bounded. The lower bound assumption of density guarantees the regularity of the optimal transport map. It is also a natural assumption to guarantee a sufficient amount of data everywhere on the manifold.

---

> > ### Comment · Reviewer_P4YA · 2022-08-03
> > **Reply**
> >
> > Thanks for your reply. I do agree that the theoretical results stand on their own and do not exactly require experimental justification, but my thinking was simply that providing some basic experiments (such as the one from Weakness #1) could help ground some of the theory and provide more motivation. At any rate, I still think the paper is solid; my overall assessment remains unchanged.

---

### Official Review · Reviewer_Rb3G · 2022-07-17

**Rating:** 6
**Confidence:** 4
**Soundness:** 3 good
**Presentation:** 4 excellent
**Contribution:** 3 good

**Summary:**

The authors prove that feedforward ReLU networks (eq 3) can well-approximate probability distributions which are supported on a low dimensional manifold embedded in a higher dimensional Euclidean space. They compute explicit bounds on the size of a network required to achieve approximation error on the same order as the intrinsic *statistical* error between $\hat{Q}_n$, $Q$.

For a distribution $Q$ supported on a $d$-dimensional compact Riemannian manifold $\mathcal{M}$, the authors construct an oracle map $g^*$ such that $Q = (g^*)_{\\#} \rho$, where $\rho$ is a simple distribution.

Two notable facts about this construction are that (1) $\mathcal{M}$ is required to have positive *injectivity radius* and (2) the resulting $g^*$ has a form that is amenable to approximation by a ReLU NN.

As a consequence of (2), the authors are able to invoke approximation results (eg. Shen et al 2019, Ohn and Kim 2019, Yarotsky 2017) to bound the 'size' (width, depth, weight norms) of a ReLU network $g_\theta$ that well approximates $g^*$, in that $W_1( (g_\theta)_{\\#} \rho, (g^*)\_{\\#}\rho) < \epsilon$.

Combining this with existing statistical estimation results Wasserstein-1 convergence of low dimensional distributions (Weed and Bach, 2019), the authors prove explicit bounds on the size of a neural network approximation $g_\theta \approx g^*$ up to the statistical error threshold.

**Questions:**

What (if any) is the relationship between the reach of a manifold and quantities like its curvature or smoothness? Is there any intuitive reason that natural data manifolds should have positive reach?

Minor typo: line 242 “Lipscthiz” is misspelled

**Limitations:**

The main limitation of this work seems to be that its approximation-theoretic results have limited relevance to practical GAN models trained on natural data. As some examples,

- The required network size is unrealistically large.
- The result says nothing about *trained* GAN models. As the authors note, practical GANs approximate $W_1$ with an adversarial training scheme, and it is generally not known whether these schemes can accurately optimize $W_1$ (eg. Arora Zhang 2017).
- This work assumes an extremely strict version of manifold hypothesis for natural data distributions, which is a heuristic that only holds approximately in practice. Small Gaussian sampling noise breaks this assumption, for example.

However, these problems are ultimately out of the scope of the current work, which I believe makes a valuable contribution to the literature on manifold fitting with GAN models.

**Strengths And Weaknesses:**

Strengths

1. Theorem 1 generalizes over existing work by removing the need to assume that $\mathcal{M}$ is globally homeomorhpic to a Euclidean space. Instead, $g^*$ is constructed as a weighted average of transport maps corresponding to local charts. In my opinion, this construction is the main contribution of this work.
2. The proof of Theorem 1 is straightforward, intuitive, and clearly explained in the paper. To the best of my knowledge it is an original contribution that is of interest to a growing body of work on the expressivity and sample complexity of GAN models.
3. The bounds in this work require relatively mild assumptions on Q (A.C. with upper and lower bounded density; supported on a manifold with positive reach). In particular, $\mathcal{M}$ is not required to be smooth.
4. The paper is clearly written with no major typos. Proofs are correct to the best of my knowledge.

Weaknesses

1. The assumption of positive reach/positive injectivity radius isn’t justified in the paper, and it’s not intuitive whether this assumption holds for natural data. However, this assumption seems to be standard in previous work.
2. The network sizes prescribed by Theorem 1 are unrealistically large. For example, practical GAN models are not trained at width exponentially larger than input dimension.

---

> ### Author Response · Authors · 2022-08-02
> **Author Rebuttal**
>
> Thank you for your review. Here are our responses.
>
> ## Weaknesses
> 1. **The assumption of positive reach/positive injectivity radius isn’t justified in the paper, and it’s not intuitive whether this assumption holds for natural data. However, this assumption seems to be standard in previous work.**
>
>     Positive reach is a standard assumption in manifold learning literature. It is a weak assumption about the curvature of manifolds. Manifolds with positive reach are plentiful, e.g. all compact $C^2$ submanifolds isometrically embedded in $\mathbb{R}^D$ have positive reach, see Proposition 14 of [this paper on reach](https://www.utgjiu.ro/math/sma/v03/p07.pdf). Furthermore, having a lower bound on reach is a crucial condition for manifold learning. It has been shown in Theorem 1 of [this paper on manifold estimation](https://arxiv.org/abs/1705.00989) that, if we consider a class of manifolds which can have arbitrarily small reach, the tangent space estimation of this class of manifolds is not consistent (the estimation error does not converge to zero as the sample size increases to $\infty$).
>
> 2. **The network sizes prescribed by Theorem 1 are unrealistically large. For example, practical GAN models are not trained at width exponentially larger than input dimension.**
>
>     We want to clarify that the width of our network is not exponential in the ambient (data) dimension but rather exponential in the intrinsic dimension of the manifold, which we expect to be many orders of magnitude smaller. Meanwhile, exponential width is common in the neural network approximation theory literature, e.g. see [this paper on neural network approximation](https://arxiv.org/abs/1804.10306).
>
>
> ## Questions
> 1. **What (if any) is the relationship between the reach of a manifold and quantities like its curvature or smoothness? Is there any intuitive reason that natural data manifolds should have positive reach?**
>
>     The reach of a manifold can control both the curvature of the manifold and how close the manifold is to self-intersect. For example, the sectional curvature of a manifold is upper bounded by $\frac{1}{\tau^2}$. See Proposition A.1 of [this paper on reach estimation](https://arxiv.org/pdf/1705.04565) for further details. All compact $C^2$ submanifolds isometrically embedded in $\mathbb{R}^D$ have positive reach. In practice, natural data may lie on a compact $C^2$ manifold in $\mathbb{R}^D$, such as data on a smooth trajectory of a dynamical system.
>
> ## Limitations
> 1. **The required network size is unrealistically large.**
>
>     Please see our response to Weakness #2.
>
> 2. **The result says nothing about trained GAN models. As the authors note, practical GANs approximate  with an adversarial training scheme, and it is generally not known whether these schemes can accurately optimize  (eg. Arora Zhang 2017).**
>
>     We agree that the training of GANs is an important question that is outside the scope of this paper; we leave this topic for future work.
>
> 3. **This work assumes an extremely strict version of manifold hypothesis for natural data distributions, which is a heuristic that only holds approximately in practice. Small Gaussian sampling noise breaks this assumption, for example.**
>
>     Thank you for the interesting comment. It is meaningful to consider a noisy version of the manifold. However, high dimensional gaussian noise can destroy the underlying manifold structure. It is shown in Theorem 3 of [this paper on manifold estimation](https://arxiv.org/abs/1906.05014) that the tangential component of noise can lead to inconsistency of the manifold estimation. We will consider some special noise, such as normal noise, in our future work.

---

### Meta-Review · Area_Chair_G8HX · 2022-08-25

**Recommendation:** Accept
**Confidence:** Certain

**Metareview:**

This paper presents a theory of generative neural networks when it is assumed that the real-world data studied lies on a low-dimensional manifold. In particular, they prove that feedforward ReLU networks can approximate probability distributions that are supported on a low dimensional manifold embedded in a higher dimensional Euclidean space, a common assumption in machine learning.

The clear consensus among reviewers is that this constitutes a valuable (and mathematically impressive) contribution to the theoretical literature on manifold fitting with GAN models, giving a strong foundation to the well-known fact that networks such as GAN can approximate real data extremely well.  The authors have also clearly answered the question raised by the various reviewers.

**Award:**

No

---

### Decision · Program_Chairs · 2022-09-14

Accept